He *et al. Genome Biology* (2020) 21:235

**METHOD**

# sn-spMF: matrix factorization informs tissue-specific genetic regulation of gene expression

Yuan He[1], Surya B. Chhetri[2,3], Marios Arvanitis[1,4], Kaushik Srinivasan[5], François Aguet[6], Kristin G. Ardlie[6], Alvaro N. Barbeira[7], Rodrigo Bonazzola[7], Hae Kyung Im[7], GTEx Consortium, Christopher D. Brown[8*] and Alexis Battle[1,5*]

*Correspondence:
chrbro@pennmedicine.upenn.edu;
ajbattle@jhu.edu
Please find the full list of authors in GTEx in Additional file 3
[8]Department of Genetics, Perelman School of Medicine, University of Pennsylvania, Philadelphia, PA, 19104, USA
[1]Department of Biomedical Engineering, Johns Hopkins University, Baltimore, MD, 21218, USA
Full list of author information is available at the end of the article

## Abstract

Genetic regulation of gene expression, revealed by expression quantitative trait loci (eQTLs), exhibits complex patterns of tissue-specific effects. Characterization of these patterns may allow us to better understand mechanisms of gene regulation and disease etiology. We develop a constrained matrix factorization model, sn-spMF, to learn patterns of tissue-sharing and apply it to 49 human tissues from the Genotype-Tissue Expression (GTEx) project. The learned factors reflect tissues with known biological similarity and identify transcription factors that may mediate tissue-specific effects. sn-spMF, available at https://github.com/heyuan7676/ts_eQTLs, can be applied to learn biologically interpretable patterns of eQTL tissue-specificity and generate testable mechanistic hypotheses.

**Keywords:** Matrix factorization, Ubiquitous eQTLs, Tissue-specific eQTLs, Transcription factors

## Background

Understanding the genetic effects on gene expression is essential to characterizing the gene regulatory landscape and provides insights into the molecular basis of phenotypes. Expression quantitative trait locus (eQTL) studies using genotype and gene expression data have demonstrated that the genetic regulation of gene expression is pervasive ([1–5], the GTEx Consortium 2020, in submission). Additionally, numerous studies have leveraged eQTLs to characterize the molecular basis of complex phenotypic variation [6–10].

Tissues in the human body carry out universal cellular processes in addition to performing highly specialized functions, driven in large part by patterns of gene expression in each cell type [11, 12]. Characterizing the tissue-sharing and tissue-specificity of genetic effects on gene expression is therefore critical to understanding how genetic variation

leads to phenotypic changes. Recent work has identified eQTLs across a broad range of human tissues. The Genotype-Tissue Expression (GTEx) project has collected eQTL data across 49 human tissues (Additional file 1: Figure S1), which provide an unprecedented opportunity to uncover the ubiquitous and tissue-specific patterns of genetic regulation of gene expression [1].

Several methods have been developed to capture the underlying tissue-specific architecture in eQTLs across tissues. The simplest such method is based on the effect sizes or $P$ values of eQTLs to identify eQTLs specific to individual tissues or cell types [13, 14]. Such heuristic methods are computationally efficient, but require manual selection of numerous subjective thresholds that affect the interpretation of results. Statistical frameworks have been developed to jointly analyze eQTLs from different datasets, such as eQTL-BMA and Meta-Tissue [15, 16]. These methods are more computationally demanding but potentially more accurate in their estimation of tissue-specificity. However, neither class of methods addresses the underlying similarity of multiple tissues or conditions in datasets such as GTEx, which may arise from shared mechanism.

Genetic effects on gene expression are often shared across some, but not all, tissues. When defining tissue-specific patterns of eQTL effects, three issues need to be considered. First, patterns of shared effects across tissues are often not obvious a priori. Manually identifying relevant groupings of tissues or contexts is not always obvious or feasible. Second, these groupings are not necessarily mutually exclusive. A single tissue may naturally belong to two or more groups based on shared biology with both. Third, an eQTL may have effects in more than one group of tissues. For example, in GTEx, different regions of the brain often have shared eQTL effects. However, effects in cerebellar tissues sometimes align with the other brain regions, but are sometimes quite distinct. Similarly, while many eQTL effects are shared across a set of digestive tissues (esophagus, stomach, and colon), many effects are specific to different subsets of these tissues, and it is not obvious how they would be grouped manually.

Matrix factorization is a general method for automatically decomposing data into overlapping, learned patterns, and has been successfully applied in biological domains, such as modeling gene expression for overlapping sets of co-functional genes. Matrix factorization applied to eQTL statistics offers a flexible and natural approach for identifying underlying patterns across eQTLs that may indeed better reflect biological mechanisms which likewise act across related, non-mutually exclusive subsets of tissues, conditions, or samples [17]. Recently, matrix factorization has been applied in a Bayesian setting to capture the structure of genetic regulation in human tissues; however, specific modeling choices for factorizing eQTL effects in various domains remain to be comprehensively evaluated [18]. It is further unexplored what insights into regulatory mechanism and functional consequences can be gained by evaluating these complex patterns of ubiquitous and tissue-specific eQTL effects.

In this study, we propose a constrained matrix factorization model called weighted semi-nonnegative sparse matrix factorization (sn-spMF) and apply it to analyze eQTLs across 49 human tissues from the GTEx consortium. We learn a lower-dimensional representation of eQTL effects across tissues, capturing both tissue-shared and tissue-specific patterns of eQTL activity. We leverage this atlas of ubiquitous and tissue-specific eQTLs to begin to characterize the regulatory mechanisms that underlie this specificity, and compare this approach to standard methods of identifying tissue-specific eQTLs.

We demonstrate that the ubiquitous and tissue-specific eQTLs exhibit distinct patterns of cis-regulatory element enrichment and identify specific TFs that appear to drive tissue-specific genetic effects.

## Results

### Matrix factorization of multi-tissue eQTL effects

The effect of eQTL variants on gene expression varies across tissues, as has been previously observed [1, 2, 19]. To better understand common patterns of genetic impact across tissues and to characterize the mechanisms that underlie tissue-specificity, we developed and applied a matrix factorization model called semi-nonnegative sparse matrix factorization (sn-spMF). The model overall seeks to decompose an input matrix of eQTL effect sizes in each tissue (regression parameters from a linear model for eQTL mapping) into underlying patterns of tissue-sharing and tissue-specificity. This model assumes that the effect size vector of one eQTL across tissues can be approximated as a linear combination (weighted sum) of learned "factors," where every factor is a vector representing one common pattern of eQTL effect sizes across tissues (Fig. 1a). When many entries in the factor are small or zero, as our model will enforce, a factor points to a subset of tissues that are commonly affected by the same eQTLs. Then, for a given eQTL, the loadings, or "weights," on each factor reflect how strongly that eQTL's effects are explained by that factor (and corresponding non-zero tissues). Given a multi-tissue dataset of eQTL association statistics as input, we identified a set of explanatory tissue factors by minimizing an objective function combining two components: (1) a weighted squared error term that captures how well the learned weights and factors reconstruct the observed eQTL effect sizes and (2) a regularization term that encourages sparsity, or many zero entries, in both factors and weights through an L1 penalty (Fig. 1b). Since it has previously been shown that inconsistent directions of effect for eQTLs will often arise from allelic heterogeneity rather than true sharing [20, 21], we constrained factors to be nonnegative.

By optimizing the objective function using alternating least squares applied to the GTEx v8 data across 49 tissues, we learned a factor matrix $F$ with 23 factors (see the "Methods" section, Additional file 1: Figure S1, S2). These factors can be categorized into two major types: a ubiquitous factor, which captures eQTLs with largely consistent effects across all 49 tissues, and tissue-specific factors, which reflect effects only found among subsets of individual tissues. Tissue-specific factors include two subtypes: 8 factors representing combinations of tissues and 14 factors representing single tissues. Each of the 8 multi-tissue factors involves closely related tissues. For example, factor 2 represents effects of eQTLs in 13 brain regions; factor 15 represents effects in transverse colon and small intestine. For interpretability, each factor is named based on the tissues it represents (Additional file 1: Figure S2). In total, 41 out of 49 tissues are represented by non-zero values in at least one tissue-specific factor. The 8 tissues that do not appear in any tissue-specific factor have significantly smaller sample sizes compared to the 41 tissues captured by one or more factors (two-sided $t$ test $P$ value = 0.024, Additional file 2: Table S1), and thus, fewer eQTLs are detected that are unique to those tissues.

### Identification of ubiquitous and tissue-specific eQTLs using sn-spMF

For each individual eQTL, we identified the relevant patterns of tissue-sharing and tissue-specificity by estimating the contribution from each of our learned factors to the eQTL's

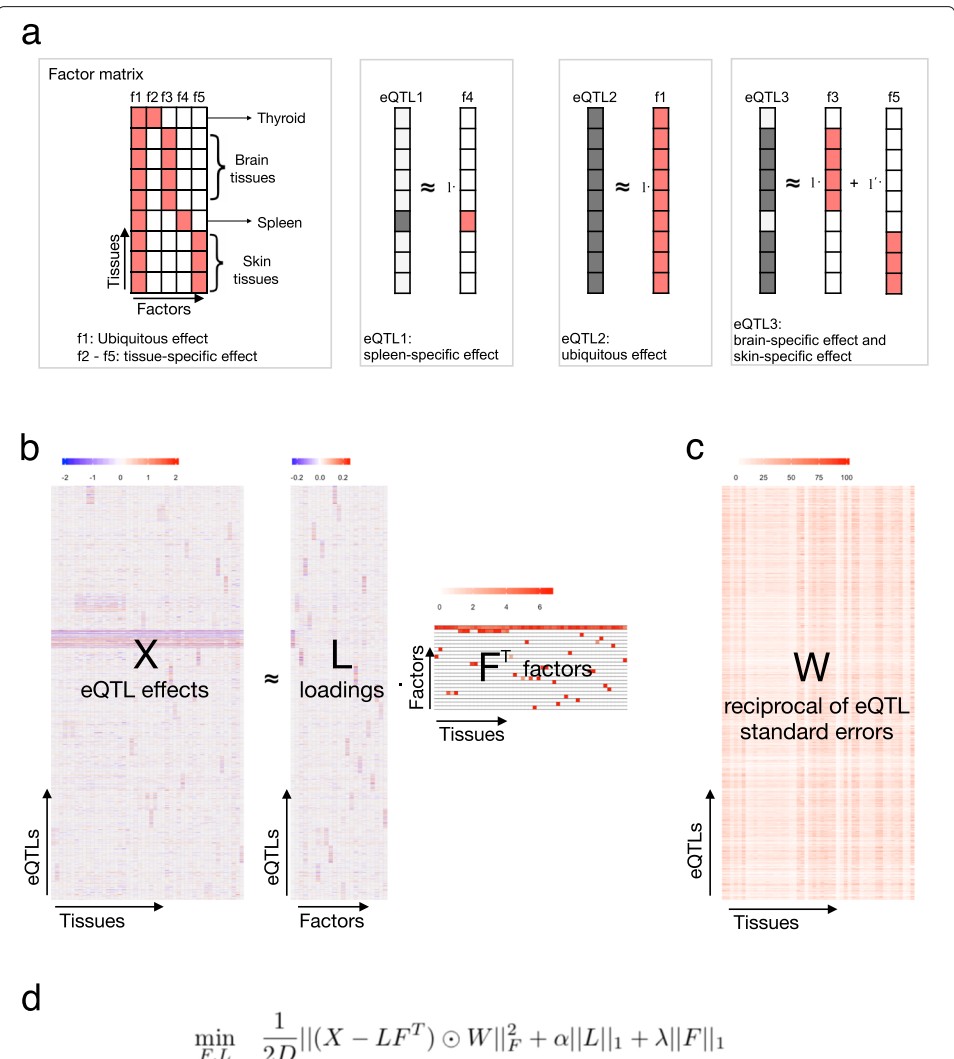

**Fig. 1** Matrix factorization model to dissect eQTL effects across tissues. **a** Simplified examples of the relationship between eQTL effect sizes and factors. eQTL1: the effect of an eQTL in the spleen can be represented by a spleen-specific factor. eQTL2: the effect of an eQTL in all nine tissues can be summarized as a ubiquitous effect across all tissues. eQTL3: the effect of an eQTL in four brain tissues and three skin tissues can be summarized as the summation of brain-specific effect and skin-specific effect. **b** Learning factors underlying eQTL effects from GTEx. *X* matrix represents the effect size of eQTLs across tissues (see the "Methods" section). Patterns of tissue-sharing and tissue-specificity are observed in *X*. Matrix factorization is implemented to learn the factor matrix *F*, where each factor captures a pattern of eQTL effect sizes across tissues. **c** Matrix *W* represents the weights for each eQTL across tissues. Each weight is the reciprocal of the standard error. **d** The objective function in sn-spMF, where $\alpha$ and $\lambda$ are sparsity penalty parameters, and *D* is the number of eQTLs

effect sizes, using a second pass of weighted linear regression (see the "Methods" section). The observed patterns of tissue-sharing and tissue-specificity and how they are decomposed by matrix factorization are illustrated in the four following examples. First, an eQTL for GLT1D1 is highly specific to the liver and loads only on the corresponding liver factor (Fig. 2a). Second, an eQTL for AATF loads on the brain tissue factor and the tibial nerve factor to explain its combined effect size profile (Fig. 2b). Although this eQTL has small effects (or large variance) in some brain subregions, the model is able to identify a brain-wide effect as a likely explanatory factor for this eQTL. Third, an eQTL for

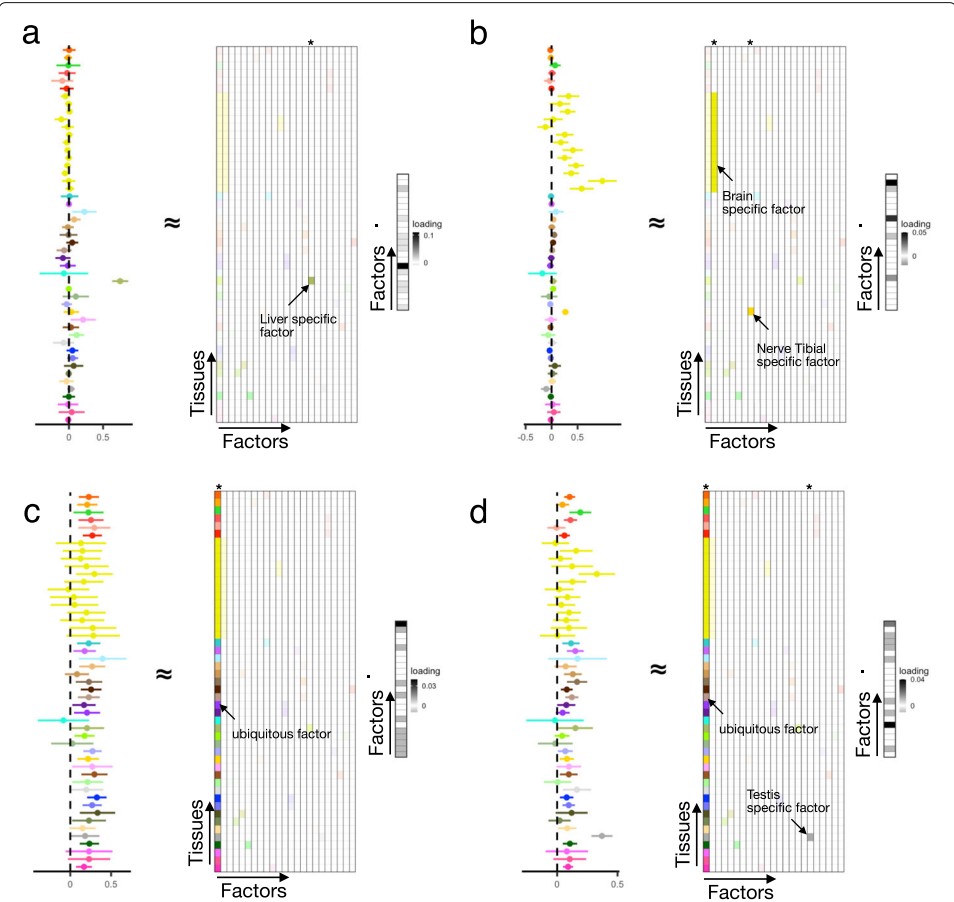

**Fig. 2** Assignment of eQTLs to factors. Effect sizes and 95% confidence intervals of four eQTLs across 49 tissues are illustrated. The fitted linear combination of factors for the eQTL is displayed in gray scale at the right of each panel. Faded colors indicate factors with coefficients with FDR ≥ 0.05. Asterisk on the tissue indicates that this eQTL was significant with FDR < 0.05 in that tissue. **a** A liver-specific eQTL (GLT1D1-rs1012994). **b** An eQTL (AATF-rs76014915) with activity in brain tissues and tibial nerve. **c** A ubiquitous eQTL (U2AF1-rs234719). **d** An eQTL (CD14-rs2563249) with ubiquitous and testis-specific effects

U2AF1 with relatively consistent effects across tissues loads only on the ubiquitous factor (Fig. 2c). Finally, an eQTL for CD14 has consistent effects across all tissues in addition to a stronger effect specific to the testis (Fig. 2d).

In summary, 1,076,761 eQTLs (20% of tested eQTLs) load on the ubiquitous factor; we refer to these eQTLs as "ubiquitous eQTLs" (u-eQTLs). For each tissue-specific factor, 76,976 to 431,585 eQTLs (1.5 to 8.1% of tested eQTLs) have significant loadings; we call these eQTLs "tissue-specific eQTLs" (ts-eQTLs) (Fig. 3a, Additional file 2: Table S2). Identified ts-eQTLs do not appear to result from genes with low levels of tissue-specific gene expression (Figure S3). In total across factors, 2,821,650 eQTLs (53% of tested eQTLs) are found to use at least one tissue-specific factor (Fig. 3b). There are 638,784 eQTLs that load on both the ubiquitous factor and tissue-specific factors (59% of the u-eQTLs and 22% of the ts-eQTLs, Fig. 3c), indicating that in addition to a broad, shared effect across tissues, these eQTLs have a much stronger effect on expression in a particular subset of tissues. eQTLs tend to load on a small set of tissue-specific factors,

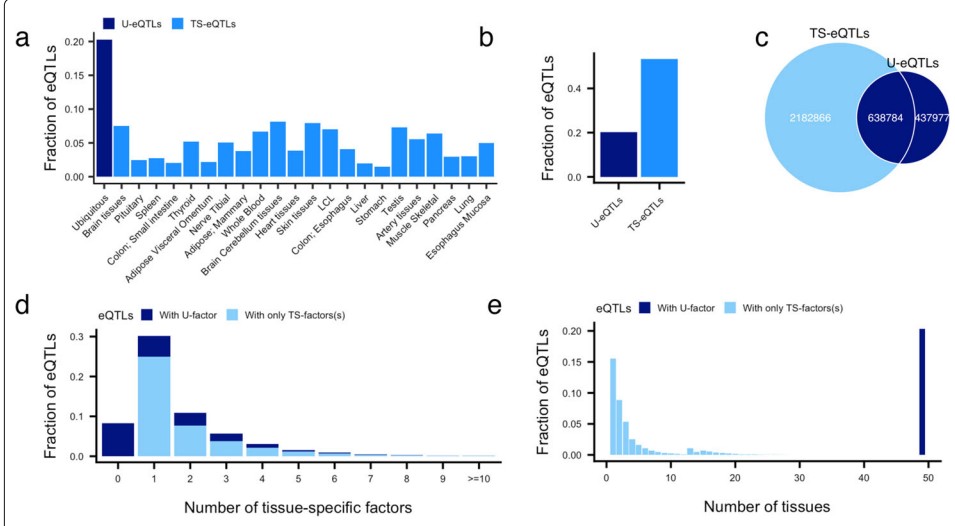

**Fig. 3** Identification of tissue-specific and ubiquitous eQTLs. **a** Fraction of tested eQTLs that load on each factor. **b** Fraction of eQTLs that load on ubiquitous and tissue-specific factors. **c** The overlap of tested eQTLs that loaded on the ubiquitous factor (u-eQTLs) and any tissue-specific factor (ts-eQTLs). **d** Fraction of eQTLs that load on different numbers of tissue-specific factors. eQTLs that load with a specific number of ts-factors can fall into one of two categories: those with the ubiquitous factor and those with only ts-factors. The figure shows the fraction of tested eQTLs that load on each number of ts-factors with colors to show the contribution for each category. **e** Fraction of eQTLs with activity in different numbers of tissues. The numbers of unique tissues represented in the set of factors for each eQTL are summed

with 3,083,103 eQTLs (99% among the eQTLs loaded on at least one factor) using less than six tissue-specific factors (Fig. 3d).

The number of factors an eQTL loads on should provide a more biologically interpretable indication of the number of independent contexts in which an eQTL is active, rather than simply counting individual significant tissues. Datasets often contain multiple similar or even duplicate tissues, such as the thirteen brain regions in GTEx, or the two skin tissues that only differ by sun exposure. It may be misleading to count a neuron-specific eQTL as active in thirteen tissues, not at all comparable to a very general eQTL active in thirteen highly distinct tissues. Here, we demonstrate that eQTLs tend to be active in just a few factors, tailing off rapidly, but these factors sometimes correspond to numerous tissues (Fig. 3d, e), providing some interpretation for the familiar "U-shape" curve that has been reported previously ([22], the GTEx Consortium 2020, in submission). However, we note that 8 tissues are not significantly represented by any tissue-specific factor and, therefore, cannot be captured in this analysis (Additional file 2: Table S1).

**Matrix factorization improves biological interpretation over heuristic methods of determining tissue relevance**

The method most commonly used to identify ts-eQTLs is simply to apply heuristic thresholds based on effect sizes, *P* values, or meta-analysis results for individual tissues [13, 14, 16, 19]. If an eQTL statistic exceeds the chosen threshold for a given tissue, and remains below another threshold for other tissues, it is considered to be tissue-specific. None of these approaches consider common patterns of tissue-sharing and may

obscure eQTL mechanisms shared across a subset of tissues (such as the brain or endothelium) unless they were manually predefined for investigation. Moreover, none of these approaches handle complex patterns of tissue-specificity, where an eQTL influences more than one tissue or predefined set, but is not universally shared.

Based on heuristic thresholds on individual tissue $P$ values (heuristic$_1$, see the "Methods" section), we identified 312,502 u-eQTLs and between 1374 and 102,414 ts-eQTLs per tissue—far fewer eQTLs are confidently assigned to each category compared to results from sn-spMF (Additional file 1: Figure S4; Additional file 2: Table S2). This difference is partly because standard heuristic methods allow only one pattern (a single tissue or a ubiquitous effect) to be assigned to each eQTL, while matrix factorization allows multiple factors and tissues to be involved in explaining the effect size of an eQTL (Additional file 1: Figure S5). In addition, heuristic methods often miss small effects from similar tissues, while matrix factorization is able to aggregate effects for similar tissues (Fig. 2). We also tried manually grouping together tissues with clear shared biology and applying heuristic thresholds based on these (heuristic$_2$, see the "Methods" section, Additional file 2: Table S3), resulting in 175,637 u-eQTLs and between 1460 and 201,584 ts-eQTLs (Additional file 1: Figure S6, S7). In subsequent sections, we show that matrix factorization allows for the identification of more biologically coherent eQTLs than heuristic approaches by comparing sn-spMF to the standard approach defined by heuristic$_1$. We also show that manually defined tissue sets as in heuristic$_2$ offer only small gains over heuristic$_1$ and do not perform as well as matrix factorization either.

### Tissue-specific eQTL gene function

To examine the functional relevance of ts-eQTL genes, we ran enrichment analysis using biological processes from the Gene Ontology (GO) project [23]. We first evaluated genes with ts-eQTLs and no u-eQTL. For sn-spMF, these eQTL genes are enriched for 546 unique GO terms at FDR < 0.05 (Additional file 1: Figure S8), and the top enriched GO terms are relevant to the corresponding tissues (Additional file 1: Figure S9, S10, S11). The ts-eQTL genes from heuristic methods, however, are less enriched in GO biological processes (at FDR < 0.05, 110 enriched for heuristic$_1$, 421 enriched for heuristic$_2$, Additional file 1: Figure S12).

After initial enrichment analysis, we used a more stringent definition of tissue-specificity to restrict the analysis to the genes most unique to each factor. For sn-spMF, we selected genes appearing in less than 6 tissue-specific factors (on average 252 genes per factor). A total of 64 unique GO terms are enriched at FDR < 0.1. The enriched GO terms are related to the matched tissue(s) of the eQTLs (Fig. 4). For example, five GO terms are enriched among liver-specific genes including four metabolic processes (for steroid, drug, uronic acid, and flavonoid) and response to xenobiotic stimulus, each relevant to liver function. For heuristic$_1$, we selected genes appearing in less than 7 tissues (on average 325 genes per tissue); for heuristic$_2$, we selected genes appearing in less than 6 subsets of tissues (on average 243 genes per subset), such that the gene sets are of comparable sizes. No GO term is enriched among these gene sets for heuristic$_1$, and one GO term is enriched for heuristic$_2$ (Additional file 1: Figure S12). These results indicate that sn-spMF is able to identify eQTL genes with biological functions relevant in the corresponding tissues more effectively than heuristic methods, even with comparably stringent definitions of tissue-specific eQTL genes providing similar numbers of genes for analysis.

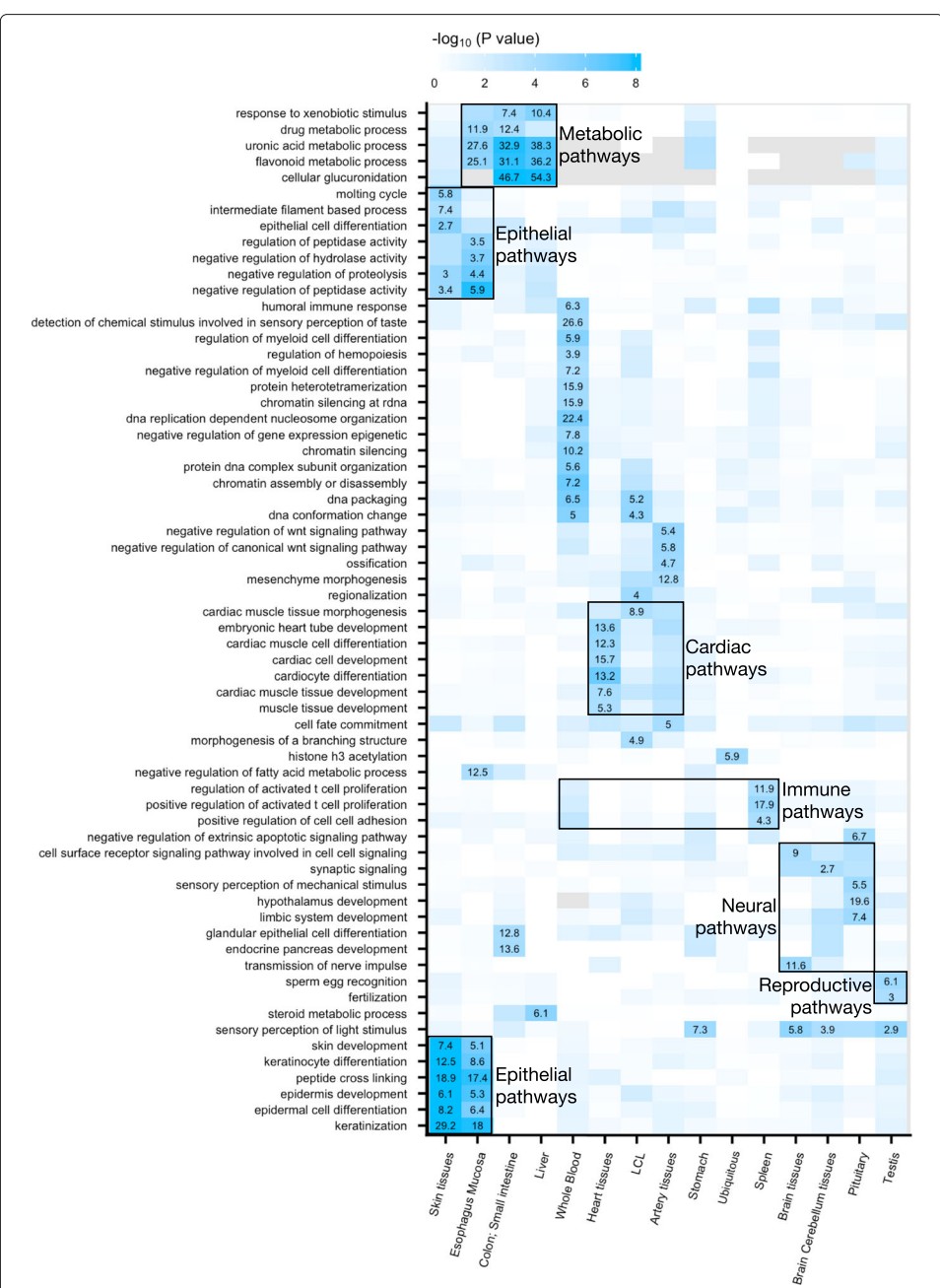

**Fig. 4** Enriched GO terms for eQTL genes from sn-spMF at FDR < 0.1. Color represents the level of enrichment ($-\log_{10} P$ value). The significantly enriched GO terms are annotated by numbers representing the odds ratio. To compute the OR for each factor, background genes include all genes tested for the represented tissues in the factor. GO terms and factors are ordered by hierarchical clustering. Examples of relevant GO terms in related tissues are annotated

### eQTL variant enrichment in cis-regulatory regions

eQTL variants are enriched in cis-regulatory elements, including cell type-specific promoters and enhancers [1, 24, 25]. Consistent with prior observations, u-eQTL variants identified by sn-spMF are more enriched in promoters (OR = 1.9, $P$ value $< 2.2 \times 10^{-16}$) than ts-eQTL variants (OR = 1.5, $P$ value $< 2.2 \times 10^{-16}$), while ts-eQTL variants are more strongly enriched in enhancers (OR = 1.3, $P$ value $= 8.5 \times 10^{-12}$) than u-eQTL

variants (OR = 1.0, *P* value = 0.40, Additional file [1]: Figure S13) [1, 26, 27]. Moreover, ts-eQTL variants are more likely than u-eQTLs to overlap enhancers whose activity is restricted to a small number of tissues (Additional file [1]: Figure S14). Compared to sn-spMF, heuristically defined ts-eQTLs exhibit comparable enrichment magnitude in enhancers (for heuristic$_1$, OR = 1.3, *P* value = $7.8 \times 10^{-8}$; for heuristic$_2$, OR = 1.4, *P* value = $4.2 \times 10^{-5}$ ), but sn-spMF provides an order of magnitude more ts-eQTLs (Additional file [1]: Figure S4, S6). While heuristic methods identify highly tissue-specific eQTLs by selecting those with effects clearly limited to a single tissue or a subset of tissues, sn-spMF identifies many more eQTLs relevant to each tissue-specific factor, each related to a shared set of cis-regulatory elements.

### eQTL enrichment in transcription factor binding sites

To systematically assess whether eQTLs for each factor are enriched in binding sites for specific TFs, we performed enrichment analysis for each of the 579 TF motifs available in the JASPAR database [28]. As a proxy for TF binding sites (TFBS) in individual tissues, we identified TF motif instances overlapping predicted enhancers and promoters [29–32].

Enrichment analysis was performed separately for TFBS in promoters and TFBS in enhancers (see the "Methods" section). In promoters, u-eQTLs and ts-eQTLs are enriched for TFBS of 136 and 181 unique TFs (median = 21 across factors), respectively (FDR < 0.05, Fig. [5]a, b). In enhancers, u-eQTLs and ts-eQTLs are enriched for TFBS of 39 and 264 unique TFs (median = 41 across factors), respectively (FDR < 0.05, Fig. [5]a, b). Among these 264 TFs, 244 (92%) are enriched for fewer than six tissue-specific factors (Fig. [5]c). Zero to 23% (among factors, median 4%) of TFs are enriched in both promoters and enhancers (Additional file [1]: Figure S15). These results indicate that ts-eQTLs are more enriched in binding sites of particular TFs in enhancers than promoters, while u-eQTLs yield more enrichment in promoters than enhancers. The heuristic$_1$ approach for identifying ts-eQTLs yields only 5 TFs enriched in promoters and 47 TFs enriched in enhancers. Similarly, there are fewer TFs enriched for heuristic u-eQTLs (59 in promoters, and 8 in enhancers, Fig. [5]a, Additional File [1]: Figure S16). Heuristic$_2$ yields 9 TFs enriched in promoters and 51 TFs enriched in enhancers for ts-eQTLs, and 97 TFs enriched in promoters and 4 TFs enriched in enhancer for u-eQTLs. The relatively low enrichment of TFBS from heuristically identified eQTLs is presumably due to the much more limited number of eQTLs identified in each category.

### Impact of matrix factorization methodological choices

In addition to our sn-spMF model, there are a variety of matrix factorization approaches available. Methodological choices include the selection of priors on loading and factor entries, which may encourage sparsity or other properties, nonnegativity constraints, and hyper-parameter selection.

We compared our method to several matrix factorization methods using simulated data (see the "Methods" section). We ran singular value decomposition (SVD) and nonnegative matrix factorization (NMF) as they are commonly used in matrix factorization. We also implemented matrix factorization with various constraints, including sparse SVD (SSVD), penalized matrix decomposition (PMD), softImpute, and nonparametric Bayesian sparse factor analysis (NBSPA) [33–36]. PMD penalizes the two decomposed matrices using either one penalty parameter scaled by the dimensions for

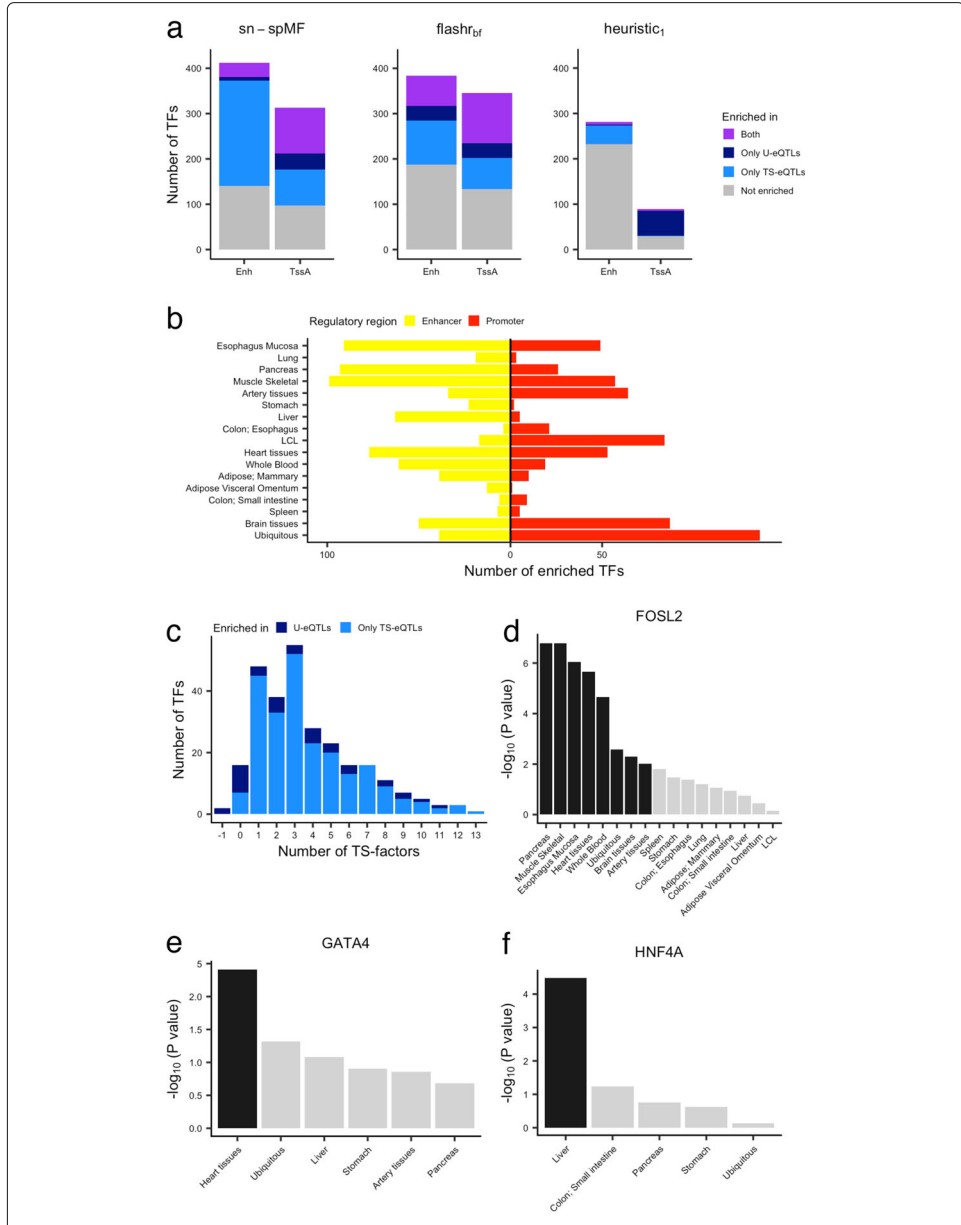

**Fig. 5** Enrichment of TFBS for u-eQTLs and ts-eQTLs. **a** Number of TFs whose binding sites are enriched for eQTLs across factors at FDR < 0.05 for sn-spMF, flashr$_{bf}$, and heuristic$_1$ methods. Enh, enhancers; TssA, active transcription start sites. **b** Total number of TFs with binding sites enriched for either only u-eQTLs, or only ts-eQTLs, or both. **c** Distribution of the number of tissue-specific factors each TF is enriched in. **d**–**f** Enrichment for example TFs among eQTLs across each factor ($-\log_{10}(P$ value) ) where the TF was expressed in corresponding tissues for **d** FOSL2, **e** GATA4, and **f** HNF4A. Black bars represent that the BH-corrected *P* value is < 0.05

each decomposed matrix (PMD$_{CV1}$) or two separate penalty parameters (PMD$_{CV2}$). Finally, we applied flashr, a recent method which uses a Bayesian framework to automatically learn the sparse structure of effects across tissues [18]. Flashr was run with default setting (flashr$_{default}$), greedily adding factors followed by backfitting (flashr$_{bf}$) and with nonnegative priors (flashr$_{NN}$). To evaluate the performance of these methods on simulated data, we computed the correlation between the learned loadings and the true loadings,

and the correlation between the learned factors and the true factors, as well as the precision and recall for true u-eQTLs and ts-eQTLs. We observed that sn-spMF and flashr$_{NN}$ achieve the most accurate loading matrix and factor matrix, and the highest precision and recall for correctly identifying u-eQLTs and ts-eQTLs (Additional file 1: Figure S17, S18), followed by other flashr approaches, NBSPA, and softImpute. Sparsity appears to confer some benefit in accuracy and interpretability of factors.

Based on strong performance in simulation, we also applied flashr methods to the GTEx data, each capturing both ubiquitous and sparse factors (Additional file 1: Figure S19). We first discuss flashr$_{bf}$, which displayed the strongest performance of the flashr methods on GTEx, in detail. Each flashr$_{bf}$ factor is somewhat more dense (more non-zero entries) than sn-spMF factors (Additional file 1: Figure S20, S21). We then identified flashr$_{bf}$ factors relevant to each eQTL using the same second pass linear regression pipeline as in sn-spMF. We thus identified 1,929,939 u-eQTLs and 69,594 to 929,009 ts-eQTLs.

Flashr$_{bf}$ ts-eQTL genes are comparably enriched for GO biological processes as sn-spMF factors, far exceeding heuristic ts-eQTL genes, with 593 enriched pathways (FDR < 0.05). However, flashr$_{bf}$ eQTL variants are not strongly enriched in enhancers (OR = 1.1, Additional file 1: Figure S22). This appears to be due to the denser flashr$_{bf}$ factors not isolating tissue-specific effects from ubiquitous effects as strongly. Assessing TF enrichment, however, because analysis is restricted to variants within enhancers identified in relevant tissues, is still able to identify enrichment for 197 TFBS across flashr$_{bf}$ factors (Fig. 5a). While regulatory element enrichment appears sensitive to matrix factorization methodological choices, both versions of matrix factorization show advantages over heuristic approaches for identifying tissue-relevant eQTL genes and for identifying particular transcription factors whose binding sites are impacted by ts-eQTL variants. Finally flashr$_{bf}$, does not include nonnegativity constraints on the factors, thus complicating interpretation of latent patterns and tissue-specificity. For example, we found that factors that contain tissues with different signs do not correspond well to patterns in the actual eQTL effect sizes—only 19–35% of eQTLs that mapped to such mixed sign factors actually display opposite sign eQTL effects in the corresponding tissues (Additional file 1: Figure S23).

For thorough comparison, we also applied other matrix factorization methods including flashr with default parameter setting (flashr$_{default}$), flashr with nonnegative prior (flashr$_{NN}$), softImpute, and PMD to the GTEx dataset (see the "Methods" section, Additional file 1: Figure S24 - S29). These methods did not offer performance gains over flashr$_{bf}$ or sn-spMF (Additional file 1: Figure S20, S21, S22, S30; Additional file 2: Table S4, S5, S6, S7). In particular, flashr$_{NN}$ provided sparse, interpretable tissue factors but suffered from multicollinearity making it difficult to distinguish ts-eQTLs from u-eQTLs (Additional file 1: Figure 25, [37, 38]). Overall, we conclude that the sparsity constraint on decomposed matrices is crucial to distinguish ts-eQTLs from u-eQTLs, and that depending on optimization approach, a nonnegativity constraint on factors can be helpful in interpreting the identified patterns of tissue-specificity.

### Transcription factors enriched in u-eQTLs and ts-eQTLs

Given the limited systematic research on the consequences of genetic variation within tissue-specific TFBS, we examined the characteristics of TFBS enriched in ts-eQTLs for each factor and in u-eQTLs. We focused on the TFBS found within enhancers because

of their generally increased tissue-specific functions (Additional file 1: Figure S13, S14). Binding sites for TFs with broad activity are enriched for u-eQTLs, such as CCAAT/enhancer-binding proteins (CEBPB, CEBPD, CEBPG), T-box 1 (TBX1), and AP-1 Transcription Factor Subunit FOSL2 [39–42] (Fig. 5d). The enrichment of these TFBS in u-eQTLs reflects their participation in a wide range of regulatory processes across tissues.

The enrichment of binding sites for 264 TFs in ts-eQTLs demonstrates their role in regulating gene expression in particular subsets of tissues corresponding to each factor. Among these, binding sites for 172 TFs display enrichment in ts-eQTLs for multiple factors with biologically plausible patterns across tissue groups. For example, hepatic nuclear factor HNF1A, known to be crucial for the development and function of the liver, pancreas, and gut epithelium, are enriched for the liver-specific eQTLs, pancreas-specific eQTLs, and ts-eQTLs for a factor reflecting the colon and small intestine [43, 44]. Furthermore, 92 TFBS are enriched in ts-eQTLs for one tissue-specific factor. Examples include binding sites for the well-characterized cardiac TF GATA4, which are enriched for heart-specific eQTLs [45, 46] (Fig. 5e); hepatocyte nuclear factor HNF4A, which are enriched for liver-specific eQTLs [47, 48] (Fig. 5f); and myogenic factor 4 MYOG, which are enriched for skeletal muscle-specific eQTLs [49] (Additional file 1: Figure S31). We continue to explore two TFs in more detail in the following sections. More examples of enriched TFs with previously characterized tissue-specific functions can be found in Additional file 1: Figure S31 and Additional file 2: Table S8.

### Heart-specific eQTLs are enriched in GATA4 binding sites

Previous studies have demonstrated the essential roles of GATA4 in heart morphogenesis [50]. In mouse studies, GATA4 has been shown to recruit the histone acetyltransferase p300 in a tissue-specific manner in the heart [45]. This GATA4-p300 complex deposits H3K27ac at cardiac enhancers, thus stimulating transcription of genes necessary for heart development. In human, missense mutations in GATA4 are associated with multiple heart diseases such as cardiac septal defects and cardiomyopathy [51, 52]. However, common genetic variants affecting GATA4 TFBS have not previously been shown to be enriched for effects on expression in cardiac tissues. Binding sites of GATA4 in heart enhancers are enriched for heart-specific eQTLs (OR = 1.7, $P$ value = 0.004, Fig. 5e), highlighting the importance of GATA4 in normal physiological conditions of the heart. Among the 48 genes loading on the heart-specific eQTL factor with variants located in TFBS of GATA4, we note that STAT3 has been reported to exhibit a crucial role in cardiomyocyte resistance to physiological stress stimuli [53].

### Liver-specific eQTLs are enriched in HNF4A binding sites

Variants in liver-specific HNF4A binding sites are enriched for eQTLs loading on the liver-specific factor (OR = 2.9, $P$ value = $3.3 \times 10^{-5}$, Fig. 5f). The enrichment of HN4FA binding sites has not been previously identified among liver eQTLs. HNF4A is an essential TF during liver organogenesis and development [47, 48] and harbors a missense mutation (rs1800961) strongly associated with liver relevant traits including high-density lipoprotein levels and total cholesterol [55–57] (Additional file 1: Figure S32).

With the availability of Chromatin Immunoprecipitation followed by high-throughput Sequencing (ChIP-seq) data for HNF4A in human liver tissues in ENCODE, we are able to directly map the genome-wide binding sites of HNF4A. Replicating the motif-based

enrichment described above, liver-specific eQTLs are strongly enriched in HNF4A ChIP-seq peaks (OR = 3.6, $P$ value $< 2.2 \times 10^{-16}$). The enrichment is not as strong in ts-eQTLs for other tissues (OR = 1.8 in the testis to 2.6 in the pancreas). Also, liver-specific eQTLs are significantly more enriched in HNF4A binding sites than are u-eQTLs (OR = 1.7, $P$ value $< 2.2 \times 10^{-16}$).

We hypothesized that variants in HNF4A binding sites lead to liver-specific eQTLs via differential binding of HNF4A. We quantified allele-specific binding (ASB) of HNF4A and, as a tissue-shared control, CTCF (see the "Methods" section). Liver-specific eQTLs are indeed significantly enriched for ASB of HNF4A (OR = 1.4, $P$ value = 0.003), but not CTCF (OR = 0.8, $P$ value = 0.4). This finding supports the possibility that the enrichment of liver-specific eQTLs in HNF4A motifs reflects altered binding affinity of HNF4A at these eQTL variants, providing a testable hypothesis for experimental validation.

### Example eQTL variant in HNF4A binding site relevant to liver phenotypes

Among the liver-specific eQTLs identified by sn-spMF, rs9987289 exhibits significant ASB for HNF4A (Fig. 6a,b, Additional file 1: Figure S33). The *A* allele is associated with increased HNF4A binding (ChIP-seq read ratio = 7.7, two-tailed binomial test $P$ value $= 8.8 \times 10^{-5}$) and with significantly lower expression of the eGene TNKS (Fig. 6b, c). HNF4A may act as a repressor of TNKS, and these data suggest that the *A* allele of rs9987289 may act by increasing binding of HNF4A and therefore reducing expression levels of TNKS. Though HNF4A has been widely reported as a transcriptional activator, it has also been associated with transcriptional repression [58–62] (Fig. 6d). Rs9987289 is located in a flanking active promoter (TssAFlank) region surrounded by enhancers in liver, while it is found in quiescent or heterochromatin regions in all 13 non-liver tissues where HNF4A is expressed (Additional file 1: Figure S34, S35).

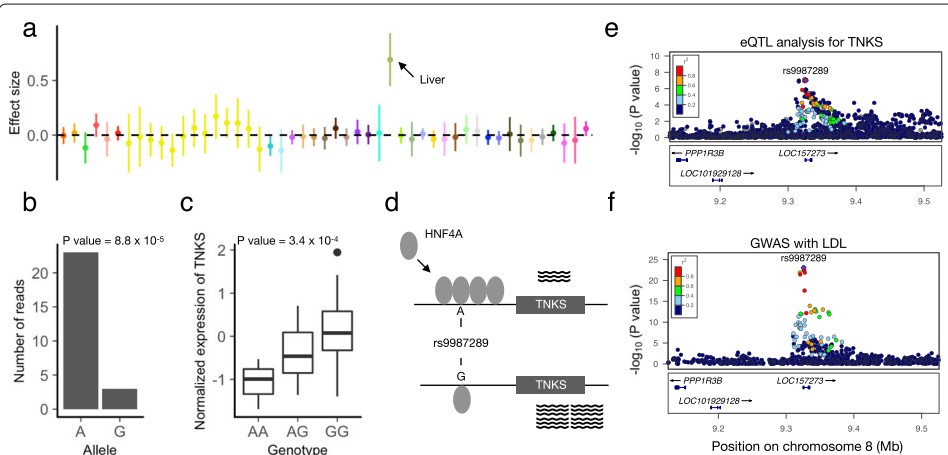

**Fig. 6** Example liver-specific eQTL, TNKS-rs9987289, in a TFBS of HNF4A that co-localizes with liver-specific phenotypes. **a** Effect size and 95% confidence interval of TNKS-rs9987289 across 49 tissues in GTEx. **b** Allele-specific HNF4A ChIP-seq reads over rs9987289 in the liver (see the "Methods" section, two-sided binomial test $P$ value $= 8.8 \times 10^{-5}$). **c** Normalized expression levels of TNKS in the liver among individuals with different genotypes at rs9987289. $P$ value $= 3.4 \times 10^{-4}$ from GTEx eQTL analysis. **d** Schematic illustration of hypothesized mechanism: allele-specific binding of HNF4A at rs9987289 and altered levels of expression of TNKS. **e** Manhattan plot (LocusZoom v0.4.8) [54] of TNKS expression levels in the liver around rs9987289. **f** Manhattan plot for LDL GWAS around rs9987289

Furthermore, rs9987289 is significantly associated with several liver-related phenotypes, including low-density lipoprotein (LDL) cholesterol levels and high-density lipoprotein (HDL) cholesterol levels [REF GTEx GWAS companion] [55] (Additional file 1: Figure S36). The liver eQTL of TNKS and the association statistics for LDL are strongly co-localized (posterior probability of shared causal signal between LDL and the eQTL = 0.94, with rs9987289 having the highest posterior of being the shared causal variant) [63] (Fig. 6e, f). Though TNKS has been widely recognized for its role in controlling telomere length, there is emerging evidence of TNKS participating in liver metabolism [64, 65].

Together, these results support the hypothesis that the tissue-specific regulatory effect of ts-eQTL variant rs9987289 in the liver may have phenotypic consequences: an active cis-regulatory element unique to the liver, allele-specific binding of liver TF HNF4A in hepatocytes, and finally co-localization of the eQTL effect with lipid GWAS hit. Such examples can provide testable hypotheses regarding multiple steps of the mechanism through which genetic variation may affect a high-level phenotype.

## Discussion and conclusions

In this study, we explored the genomic context and potential mechanisms underlying tissue-specific effects of genetic variation by applying a constrained matrix factorization model (sn-spMF) to multi-tissue eQTL data from the GTEx project. Using sn-spMF, we learned factors representing the common patterns of eQTL sharing across tissues, such as factors corresponding to ubiquitous effects across all tissues and effects shared among only brain tissues or among muscle tissues. This allowed us to explore eQTL effects shared across overlapping subsets of tissues that share cis-regulatory mechanisms due to shared cell types or developmental origin, without having to manually prespecify each such pattern. These learned factors enabled us to evaluate potential mechanisms relevant to genetic effects following these patterns of tissue-sharing.

sn-spMF identified much larger sets of tissue-specific eQTLs than did heuristic methods. The ts-eQTLs from sn-spMF were also equally or more enriched for GO biological processes, transcription factor binding sites, and tissue-specific cis-regulatory elements than the heuristic ts-eQTLs. These results suggest that sn-spMF identifies larger numbers of ts-eQTLs that remain biologically coherent, offering an opportunity for novel mechanistic insights. Other versions of matrix factorization, such as flashr, also provide meaningful views of tissue-specificity. In particular, we note the flashr has the advantage of learning the parameters with less computational burden, compared to sn-spMF where a grid search is needed for tuning parameters.

There can be other definitions of the manually selected subsets of tissues. However, it is not clear how to choose the relevant tissues and the thresholds before we have learned the latent patterns. For example, it is not clear whether whole blood and spleen should be grouped into one factor, or used as two separate factors. Also, heuristic methods can be hard to implement in situations where we have little knowledge about the feature (in contrast to our knowledge of tissue similarity). For example, in a time-series data, it is typically unknown, a priori, how patterns change during the time course.

The large set of ts-eQTLs provided by sn-spMF enabled a detailed evaluation of eQTLs in transcription factor binding sites that was not possible from heuristic approaches.

We evaluated 76,976 to 431,585 ts-eQTLs for enrichment in promoter and enhancer elements, and were able to identify 181 and 264 TFs enriched among these, respectively. This list of 264 TFs enriched in ts-eQTL enhancers provides experimentally testable hypotheses about specific genetic variants within TFBS that alter expression in a tissue-specific fashion.

Matrix factorization is inherently limited by the eQTL data used as input to the method—any tissue that is underpowered or not well represented in the original eQTL dataset is unlikely to be captured strongly by a ts-eQTL factor with sn-spMF. Further, sn-spMF does not explicitly model linkage disequilibrium (LD) or consider allelic heterogeneity, rather it relies on the user to pre-select candidate causal variants using fine-mapping tools or other approaches. Additionally, many matrix factorization approaches, priors, and constraints remain to be explored that may capture different properties of the eQTL data than represented here. Different applications, such as time series or perturbation-response eQTL data, may ultimately benefit from specialized matrix factorization formulations [17].

In conclusion, we have developed a constrained matrix factorization model to learn patterns of eQTL tissue-specificity across 49 human tissues using data from GTEx v8. We observed improved enrichment of biologically relevant genes and cis-regulatory elements compared to heuristic methods. Matrix factorization also revealed the potential impact of ubiquitous TFs on ubiquitous eQTLs and provided a list of candidate TFs relevant to each tissue-specific set of eQTLs.

## Methods
### GTEx data
GTEx Release v8 project has collected both genotype data from whole genome sequencing (WGS) and RNA sequence (RNA-seq) from 838 people. Here, we analyze GTEx data from 15,253 samples, consisting of 47 tissues and two cell lines (the GTEx Consortium 2020, in submission). GTEx v8 data release includes cis-eQTL analyses that test for association between gene expression and variants within 1 MB of the genes' transcription start sites (TSS). Effect sizes of the eQTLs are represented by coefficients estimated in the linear model association tests.

### *Preprocessing and input data*
To restrict the analysis to potential casual variants, we used cis-eQTLs that are in the 95% credible set for at least one tissue [66]. Specifically, for each eQTL gene, the credible set consists of eQTL variants that include the causal variant with 95% probability. In total, 5,301,827 eQTLs with 17,480 unique protein coding eQTL genes are included in the analysis. For these 5,301,827 eQTLs, we collected the effect size and standard error from univariate cis-eQTL analysis across tissues, based on the linear model association test results from GTEx (the GTEx Consortium 2020, in submission). Missing entries, corresponding to tissues where an eQTL variant-gene pair was not tested, were assigned weights of 0 and thus do not contribute to the objective function of sn-spMF. This avoids biasing towards shared eQTLs caused by removing data points with any missing data. Finally, the lead variants, within credible sets, with the most extreme geometric mean $P$ values across tissues for the 17,480 eQTL genes were used as input (rows in matrix $X$ and $W$) to learn the factor matrix (matrix $F$). Ultimately, only 17,480 of the original 5,301,827

eQTLs are used to learn the factor matrix. However, the learned sn-spMF representation can then be used to analyze any tested eQTL variant.

sn-spMF is able to learn the underlying patterns from a subset of representative eQTL summary statistics. In our case, we restricted to credible set variants with the strongest signals across tissues, as described above. Other users may choose another representative subset of variants of interest based on their preferred methods for selecting likely causal variants or lead variants, but regardless, sn-spMF does not require summary statistics for every tested variant to learn relevant factors.

### Lower-dimensional representation of eQTL effects

eQTL effects across tissues can be represented by a matrix $X_{D \times T}$ where $D$ is the number of eQTLs and $T$ is the number of tissues. Each entry is the regression parameter obtained from eQTL association testing of one variant/gene pair in one tissue, in the case of GTEx based on a standard linear model. Each row is then the effect of one eQTL across all tissues, and each column is the effect of all eQTLs for one tissue. The effect values are real-valued and can be positive or negative. A lower-dimensional representation of the effect matrix $X$ can be written based on a factor matrix $F_{T \times K}$ and a loading matrix $L_{D \times K}$ such that $X \approx LF^T$ (Fig. 1).

### *Weighted semi-nonnegative sparse matrix factorization algorithm sn-spMF*

In order to describe the eQTL effects, we designed a matrix factorization objective function with several features: (1) *A penalty on a weighted sum of residuals*: in order to account for uncertainty in effect size estimates, the residual for each data point was weighted by the reciprocal of its standard error. In this way, data points with more certain eQTL effect sizes have more influence over optimal parameter estimates. Missing values in the input data were assigned a weight of zero and thus do not influence the value of the objective. (2) *Sparsity*: to alleviate over-fitting, an l1 penalty was applied to the decomposed matrices. (3) *Semi-nonnegativity of the decomposed matrices*: the factors capture the pattern of effects across tissues, and thus, it was a natural constraint to make the factors nonnegative for ease of interpretation. At the same time, because the input matrix has mixed signs, there was no such constraint on the loading matrix. The objective function was formulated as below:

$$\min_{F,L} \quad \frac{1}{2D}||(X - LF^T) \odot W||_F^2 + \alpha||L||_1 + \lambda||F||_1$$

where $F$ is nonnegative, $W$ is the element-wise reciprocal of the standard error of the eQTLs, $D$ is the number of data points (in this case the number of eQTLs), and $\alpha$ and $\lambda$ are the penalty parameters.

This objective function is biconvex, that is, convex only in $F$ or in only $L$ given the other, but not convex in both jointly. We used alternating least squares (ALS) with gradient descent to optimize the objective (Algorithm 1, implemented in `R version 3.5.1`, [67, 68]). At each iteration, we fixed $F$ and updated $L$, and then fixed $L$ and updated $F$. The update was finished when the Frobenius norm of difference in $F$ between two iterations was $< 0.01$. In each update step, the optimization problem was a linear regression with constraints. Since the solution to linear regression was guaranteed to minimize the sum of mean squared error and penalty, the cost function monotonically decreased.

---

**Algorithm 1** Weighted semi-nonnegative sparse matrix factorization algorithm (sn-spMF)

1: Input: $X_{D \times T}$

2: Output: $L_{D \times K}, F_{T \times K}$

3: Randomly Initialize nonnegative $F$

4: **while** not converged **do**

5: **for** $i = 1 \ldots D$ **do**

6: $l_i \leftarrow \min_{l_i} || \left( x_i - l_i F^T \right) \odot w_i ||_F^2 + \alpha ||l_i||_1$

7: which is equivalent to

8: $l_i \leftarrow \min_{l_i} ||x_i \odot w_i - l_i \left( F^T \mathrm{diag}(w_i) \right) ||_F^2 + \alpha ||l_i||_1$

9: **end for**

10: **for** $j = 1 \ldots T$ **do**

11: $f_j \leftarrow \min_{f_j} || \left( x_j - f_j L^T \right) \odot w_j ||_F^2 + \lambda ||f_j||_1, ||f_j|| \geq 0$

12: which is equivalent to

13: $f_j \leftarrow \min_{f_j} || \left( x_j \odot w_j - f_j \left( L^T \mathrm{diag}(w_j) \right) \right) ||_F^2 + \lambda ||f_j||_1, ||f_j|| \geq 0$

14: **end for**

15: **end while**

---

### Model selection

In the sn-spMF model, we need to set hyper-parameters including the rank of the decomposition ($K$) and the sparsity penalty ($\alpha, \lambda$). We evaluated $K$ within $[20, 25, 30, 35, 40]$, and $\alpha$ and $\lambda$ within $[4.9, 24.5, 49, 245, 490]$. These ranges were chosen by considering the number of tissues in GTEx to define plausible values for $K$ and by manual inspection of solutions for widely varying $\alpha$ and $\lambda$ to avoid high-resolution search for ranges of these hyper-parameters that resulted in clearly implausible solutions, such as lack of sparsity or large numbers of empty, un-utilized factors.

Within these chosen search spaces, we evaluated sn-spMF models for all combinations of $K, \alpha$, and $\lambda$ using (1) a previously defined criterion of matrix factorization stability and (2) independence of the learned factors, which represents adequate sparsity. Considering the stochastic nature of matrix factorization, Brunet et al. proposed a method looking for the most stable factorization result, and this method has been applied in various studies [69, 70]. We obtained the consensus matrix $C$ after 30 runs with random initialization for each model. The values in $C$ are between 0 and 1, representing the proportion of runs in which a pair of tissues are assigned to the same factor. Using the $C$ matrix, we computed the cophenetic correlation which is used to measure the degree of dispersion for the $C$ matrix. Higher cophenetic correlation indicates a more stable factor matrix.

Evaluating the runs for all combinations of hyper-parameter settings, we first eliminated some settings of $K$. Here, for each observed mean number of learned, non-empty factors $K'$ (which may be less than the input $K$), we aggregated across the different settings of $\lambda$ and $\alpha$ and computed the median cophenetic correlation [69]. We eliminated from consideration any settings of $K$ corresponding to a $K'$ with a median cophenetic correlation $< 0.9$. Next, among the remaining individual settings, we eliminated any cophenetic correlation $< 0.9$. Last, among these apparently stable settings, we selected the final hyper-parameters based on the minimum Pearson correlation between pairs of factors, to encourage independent factors and a level of sparsity that

matches independent signals in the data. Here, we computed the Pearson correlation for each pair of factors, took the Frobenius norm of the pairwise correlation matrix, and averaged this across the 30 randomly initialized runs for the same setting. Documented code and examples of the model-selection process are available on Github (https://github.com/heyuan7676/ts_eQTLs)

### Assignment of eQTLs to factors

After we have learned the factors, we identify a set of relevant factors for each eQTL using weighted linear regression. Specifically, for each eQTL, a weighted linear regression of the form $x = FL$ is fit, where $x$ is the vector of eQTL effect sizes across tissues, $F$ is the factors learned from sn-spMF, and $L$ are the regression coefficients. Weights $w$ are incorporated, where $w_t$ is the reciprocal of the standard error for the eQTL effect size $x_t$ in tissue $t$. Weighted linear regression using standard error in this manner is a common approach allowing data points with high uncertainty to have less influence on the regression parameter estimates [71]. Statistical significance of each factor for the eQTL is determined according to $P$ values based on the standard $t$ test from this linear regression. To alleviate the multiple testing burden, we removed the eQTLs for which the variants were in perfect LD ($R^2 = 1$) with variants from another eQTL before running regression for the remaining 3,601,800 eQTLs [72]. We applied the Benjamini-Hochberg correction to get the $q$ value for every factor for each eQTL [73]. We then mapped the $q$ value back to all 5,301,827 eQTLs where the SNPs are in an LD block with the tested SNPs for the same gene. We observed that occasionally, there were factors assigned negative regression coefficients when the actual observed effect sizes in the corresponding tissues were positive, or vice versa. This discrepancy arose due to collinearity between the factors, and in such cases, the discrepant factors were not included for downstream analysis. We also removed those factors that caused one tissue to have an oppositely signed small effect (absolute $Z$-score $< 3$, or $P$ value $> 0.00135$) when compared to the factor where this eQTL has the strongest effect; such discrepancies may often reflect allelic heterogeneity or LD contamination rather than true opposite effects from the same causal variant [20, 21]

### Background SNP-gene pairs

For enrichment analyses, random SNP-gene pairs were sampled from all SNP-gene pairs to match for eQTLs by three criteria: (1) SNP MAF was matched to the eQTL variants' MAF, (2) distance from the SNP to transcription start sites (TSS) of the gene was matched to eQTL, and (3) a number of SNPs per gene were matched as in eQTLs.

### Enrichment analysis of chromatin states

For each 5 bp window centered on each SNP, we identified overlapping (1) chromatin state predictions from the Roadmap Epigenomics project and (2) regions of open chromatin identified by DNAse-seq from ENCODE [29, 30, 74–76]. In Roadmap, chromatin states are predicted for each tissue or cell type that include enhancers, promoters, and transcribed regions. We used the standard 15-state Roadmap segmentations independently for each of the samples that were matched to GTEx tissues (Additional file 2: Table S9, S4). If a tissue had more than one dataset available, we merged the datasets using BEDTools [77]. For the datasets using genome assembly hg19, we used liftOver to map the peaks

to GRCh38 [78]. We built the $2 \times 2$ contingency table for eQTLs from each factor and across the 15 chromatin states. In the table, the first row includes eQTL variants in the factor, and the second row includes randomly matched SNPs. The columns indicate the number of SNPs that are located in the tested chromatin state in the tested tissues. Both tissues matched for the factor and tissues not matched for the factor were tested. We then ran a one-sided Fisher's exact test for each contingency table and corrected the $P$ values using BH-correction. To summarize the results across tissues and across factors, we used a random-effects model (`rma()` in R) to obtain the combined odds ratio and combined standard error [79].

### Heuristic thresholding methods to derive u-eQTLs and ts-eQTLs

*heuristic*$_1$: We defined ts-eQTLs in one tissue as those with $P$ value $> 0.001$ in at least 44 other tissues, and with $P$ value $< 100\times$ the most extreme $P$ value of the eGene in the tissue of interest, and within the credible set for that tissue. The thresholds were chosen such that we have a reasonable number of ts-eQTLs, and at the same time only eQTLs with a strong effect in the tissue of interest. u-eQTLs were restricted to those found in the credible sets for at least 5 tissues.

*heuristic*$_2$: Here, we defined ts-eQTLs in manually defined subsets of similar tissues (Additional file 2: Table S3). For each subset of $N_k$ tissues, the ts-eQTLs were defined as those with $P$ value $> 0.001$ in at least $49 - N_k - 5$ other tissues, and with $P$ value $< 100\times$ the most extreme $P$ value of the eGene in $\geq 50\%$ of the tissues in the subset. U-eQTLs were restricted to those found in at least 5 different subsets of tissues.

### Simulation

We simulated data with $N = 100$ eQTLs, $T = 10$ tissues, and $K = 5$ factors with sparse loadings and nonnegative factors including a dense factor and four sparse factors. Non-zero values in the loadings were randomly drawn from a standard normal distribution. An error matrix $E$ added noise to the input matrix such that $X = LF^T + E$. Values in $E$ were randomly drawn from normal distribution with mean 0 and different levels of variance $\sigma^2$ ($\sigma^2 = 0.001, 0.01, 0.05, 0.1$). To evaluate the performance of multiple methods, we computed the correlation between the learned loadings/factors and the true simulated loadings/factors (factor orderings were permuted to reach the highest correlation for each method), and the relative root mean squared error: $RRMSE(\hat{X}, X) = \sqrt{\frac{\sum_{i,j}(\hat{X}_{i,j} - X_{i,j})^2}{\sum_{i,j} X_{i,j}^2}}$ [18].

### Other matrix factorization methods

We ran singular value decomposition (SVD) using the R function `prcomp`, and nonnegative matrix factorization (NMF) using the R package `NMF` [80]. We ran sparse SVD (SSVD) using the R package `ssvd` [33, 81], penalized matrix decomposition (PMD) using the R package `PMA` [34, 82], and softImpute using the R package `softImpute` [35, 83]. We ran flashr using the R package `flashr` [18, 84].

SSVD is reported to be robust to tuning parameters, so we ran SSVD with the default settings [18, 33]. PMD penalizes the two decomposed matrices using either one penalty parameter scaled by the dimensions for each decomposed matrix (PMD$_{CV1}$) or two separate penalty parameters (PMD$_{CV2}$). We chose the tuning parameter by cross-validation, in both PMD$_{CV1}$ and PMD$_{CV2}$ [34]. softImpute has one parameter $\lambda$, and we chose it

such that the factor matrix reaches the highest sparsity while preserving the rank [35]. To run default flashr, we ran *flashr*. To run flashr$_{bf}$, we initialized the rank 1 factor and loading using *flashr* ::: *udv$_{si}$* where the initial decomposition was done using softImpute (with penalty parameter $\lambda = 0$, [18, 83]). We then did a two-round fitting by first greedily adding factors (*flash_greedy_workhorse*) and then applying backfit (*flash_backfit_workhorse*). In flashr$_{NN}$, initialization was also done using *flashr* ::: *udv$_{si}$*, and nonnegative priors were imposed by setting $ebnm_{param} = list(l = list(mixcompdist = \text{``}normal\text{''}, optmethod = \text{``}mixSQP\text{''}), f = list(mixcompdist = \text{``}normal\text{''}, optmethod = \text{``}mixSQP\text{''}))$ [18].

### Enrichment analysis of transcription factor binding sites

To examine the enrichment of TF binding sites in u-eQTLs and in ts-eQTLs, we constructed the $2 \times 2$ contingency tables across factors for each TF. For each TF, we first annotated its binding sites by overlapping tissue-specific enhancer predictions from Roadmap Epigenomics and its TFBS predictions on the genome from JASPAR [28–30]. We then restricted analysis to genes with at least one variant located in TFBS to avoid genes intrinsically lacking variants in TFBS. In the contingency table for each TF, the first row includes eQTLs, and the second row includes randomly matched SNP-gene pairs. For u-eQTLs, the columns indicate the number of genes with or without ubiquitous variants in the TFBS. For ts-eQTLs, first column indicates the number of genes with or without tissue-specific variants in the TFBS. One thing to note is that the TFBS were annotated using matched tissues for each factor. Fisher's exact test was performed for each of these contingency tables, and the $P$ values were corrected using Benjamini-Hochberg [73].

For eQTLs from each factor, the analysis was done for TFs with median TPM $> 1$ in at least half of the corresponding tissues with available data. TFs with a total number of genes in TFBS $< 10$ were removed. The tissue-specificity of the enriched TFs is unlikely to result from filtering TFs based on expression level and the number of hits (Additional file 1: Figure S37, S38).

### Identification of allele-specific binding sites using ChIP-seq data

FASTQ files from human liver samples of HNF4A and CTCF were downloaded from ENCODE web portal and aligned to the GRCh38 genome assembly using STAR [85] (Additional file 2: Table S11). Reads that mapped to variants in GTEx and passed WASP filters were extracted [86]. BAM files of the samples and controls from the same ENCODE repository were downloaded, and peak-calling was performed using MACS2 [87]. Only reads that mapped to peaks at $q$ value $< 0.1$ were included, and ASB was computed for each variant with more than 10 reads by examining if the numbers of reads at each allele were significantly different, using a two-tailed binomial test. Variants with significant ASB events were called at FDR $< 0.05$ using Benjamini-Hochberg [73].

### Supplementary information

---

**Additional file 1:** Supplementary figure. Figure S1 - S38.

**Additional file 2:** Supplementary tables. Table S1 - S11.

**Additional file 3:** GTEx Consortium Information.

**Additional file 4:** Review history.

---

### Peer review information

### Acknowledgements

We thank Jessica Bonnie for editing the manuscript, and thank Princy Parsana for open source code.

### Review history

The review history is available as Additional file 4.

### Authors' contributions

AB, CDB, and YH designed and coordinated the project. YH performed the analysis. YH, AB, and CDB wrote the manuscript. SBC participated in the data analysis and critically revised the manuscript. MA participated in the analysis of GWAS hits. KS participated in running flashr. FA and KGA were responsible for the V8 data generation and cis-eQTL calling. ANB, RB, and HKI harmonized and imputed the GWAS summary statistics. AB and CDB conceived and supervised the study. All authors read and approved the final manuscript.

### Authors' information

Twitter handles: @alexisjbattle (Alexis Battle), @casey6r0wn (Christopher D. Brown), @YuanHe7 (Yuan He), @hakyim (Hae Kyung Im), @francoisaguet (François Aguet), @kaushiksrins (Kaushik Srinivasan).

### Funding

AB is supported by NIMH 1R01MH109905, NHGRI 1R01HG010480, and Searle Scholar's Program. CDB is supported by RF1AG05547701 and R01HL133218. MA is supported by T32 HL007227. FA and KGA are supported by GTEx program grants HHSN268201000029C, 5U41HG009494. HKI is supported by R01MH107666 and P30DK020595. The Genotype-Tissue Expression (GTEx) project was supported by the Common Fund of the Office of the Director of the National Institutes of Health (NIH). Additional funds from the National Cancer Institute; National Human Genome Research Institute (NHGRI); National Heart, Lung, and Blood Institute; National Institute on Drug Abuse; National Institute of Mental Health; and National Institute of Neurological Disorders and Stroke. Donors were enrolled at Biospecimen Source Sites funded by Leidos Biomedical, Inc. (Leidos) subcontracts to the National Disease Research Interchange (10XS170) and Roswell Park Cancer Institute (10XS171). The Laboratory, Data Analysis and Coordinating Center (LDACC) was funded through a contract (HHSN268201000029C) to The Broad Institute, Inc. Biorepository operations were funded through a Leidos subcontract to Van Andel Institute (10ST1035). Additional data repository and project management provided by Leidos (HHSN261200800001E).

### Availability of data and materials

The data and analyses presented in the current publication are based on the use of study data downloaded from the dbGaP website under phs000424.v8.p2 [88] and on the GTEx portal (http://gtexportal.org/). All the code used for the matrix factorization and mapping eQTLs is available, under the Creative Commons Attribution 4.0 International License, on Zenodo with the access code DOI: (https://doi.org/10.5281/zenodo.3969649) [89], and GitHub (https://github.com/heyuan7676/ts_eQTLs) [90]. Details of the license can be found here: http://creativecommons.org/licenses/by/4.0/. Data from ROADMAP and ENCODE project used in the analysis are listed in Additional file 2: Table S9, S10, S11.

### Ethics approval and consent to participate

Not applicable. The raw data is published with the GTEx Consortium 2020, in submission.

### Consent for publication

Not applicable

### Competing interests

FA is an inventor on a patent application related to TensorQTL; HKI has received speaker honoraria from GSK and AbbVie.

### Author details

[1]Department of Biomedical Engineering, Johns Hopkins University, Baltimore, MD, 21218, USA. [2]HudsonAlpha Institute for Biotechnology, Huntsville, AL, 35806, USA. [3]Current Address: Department of Biomedical Engineering, Johns Hopkins University, Baltimore, MD, 21218, USA. [4]Department of Medicine, Division of Cardiology, Johns Hopkins University, Baltimore, MD, 21287, USA. [5]Department of Computer Science, Johns Hopkins University, Baltimore, MD, 21218, USA. [6]The Broad Institute of MIT and Harvard, Cambridge, MA, USA. [7]Section of Genetic Medicine, Department of Medicine, The University of Chicago, Chicago, IL, USA. [8]Department of Genetics, Perelman School of Medicine, University of Pennsylvania, Philadelphia, PA, 19104, USA.

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

## 
