## [**Additional file 4** Review history. · Genome Biology]

Review History

(Text in black represents comments from the reviewers, and text in blue indicates response to the comments.)

First round of review

Reviewer #1: This manuscript reports the usage of semi-nonnegative sparse matrix factorization (sn-spMF) approach to analyze the tissue specificity of multiple eQTLs and performed the systematic analysis on the tissue-specific pattern of the eQTLs. This is an interesting study. The methods and results are well described. However, this reviewer has some technical concerns on the manuscript.

Thank you for your interest in our manuscript and for your comments and suggestions. We address specific concerns below, and note we have added numbering of your comments for easy reference, but maintained each of your original points.

1.

i). It seems to this reviewer that a major problem is the suitability of the semi-nonnegative sparse matrix factorization (sn-spMF) for dissecting tissue-specific patterns of eQTLs. It is straightforward to test whether an eQTL or multiple eQTLs are specific to a tissue.

We appreciate the importance of this concern. However, in our experience with this problem as part of GTEx and several other eQTL projects, it is actually not straightforward to assess whether an eQTL is specific to a tissue (or other condition). Making this judgement involves deciding on a series of heuristics and thresholds including A) manual groupings of tissues/conditions that are expected to share eQTLs a priori, B) lower bound cutoffs for significance in the tissue(s) of interest, C) lower bound on the fraction of target tissue(s) of interest for which the eQTL may appear, D) upper bounds on the effect size observed in “irrelevant” tissues and E) upper bounds on the number of other “irrelevant” tissues for which the eQTL may appear, all needed to decide if the eQTL is truly specific to the tissue(s) of interest. Before turning to matrix factorization, we indeed attempted to manually define tissue-specific eQTLs in many ways for this project, as we have in previous work. Especially in datasets with dozens of tissues or other overlapping groupings of samples, such as time-series, single-cell, or perturbation data, this is a frustrating exercise and results are sensitive to each choice, with no ideal setting apparent in most cases.

ii). Nonnegative matrix factorization was proposed to reduce dimension or to cluster objects. What are the advantages of sn-spMF for this specific purpose?

Matrix factorization overall offers some important conceptual benefits over heuristic approaches (and we address particular properties of sn-spMF at the end of the reply to this point).

1) The underlying patterns of tissue-sharing (analogous to “grouping” tissues) are learned from the data instead of being manually defined (Fig 1b, 1c, 1d). Manually defining the patterns can be hard in some conditions, for example, in time-series data where the similarity between time points are unknown, or in single cell data where the similarity of cell subtypes are unclear. In GTEx, some of the tissues may naturally be grouped together in multiple ways and a single, mutually exclusive grouping is not obvious. For example, different regions of the brain can often be grouped together. However, in the GTEx data, effects in cerebellar tissues sometimes align with the other brain regions, but are sometimes quite distinct. Similarly, there are often complex tissue grouping possibilities: should the esophagus be grouped with the stomach or colon, or should they all group together? In this analysis, the data support a factor that represents the colon and esophagus, with a distinct factor representing the stomach, highlighting the difficulty of identifying tissue groupings a priori (Additional file 1: Fig. S1, S2).

2) Matrix factorization naturally allows each eQTL to be assigned to more than one factor (Fig 1a, Fig 2), meaning the eQTL appears to be active in more than one tissue group. This is biologically intuitive and supports phenomena such as transcription factor (TF) binding sites being used in multiple (sometimes unrelated) cell types and conditions, even if the TF is not universal. We do indeed observe many eQTLs that appear to impact more than one “group” of tissues (Fig 3d, Additional file 1: Fig S4).

3) Heuristic methods require subjective choices of thresholds, for which there are few strategies to select appropriate values, as noted above. In contrast, MF approaches fit naturally into a framework of multivariate regression to identify tissue-specificity. There are well studied methods for hyperparameter selection for MF and regression, thus requiring fewer arbitrary thresholds (Brunet et al 2004, Wu et al 2016).

Within MF methods, sn-spMF is a particularly appropriate formulation for eQTL data analysis.

- 1) sn-spMF produces a sparse solution (few non-zero entries) for both factors and loadings, making it easy to interpret. One can readily identify the tissues playing a role in each factor, and the small subset of factors relevant to each eQTL.
- 2) sn-spMF can take input that includes missing values. eQTL effect sizes are indeed not available in every tissue for every gene, particularly in cases of low gene expression in some tissues or insufficient number of individuals carrying the eQTL variant.
- 3) sn-spMF allows input that includes both positive and negative values. eQTL effect sizes naturally can have either sign, and opposite signs in different tissues do indicate different effects. Taking the absolute value (as would be required by standard non-negative matrix factorization) would not be appropriate.
- 4) sn-spMF employs a semi-nonnegativity constraint of the decomposed matrices: the factors capture simply groupings of tissues with similar behavior, and thus it was a natural constraint to make the factors nonnegative for ease of interpretation.

Simpler, standard matrix factorization approaches do not offer all of these features, and sn-spMF performs better in simulation and on real data than other related approaches (as detailed in response to comment 1 - iii)).

iii). It is unclear how sn-spMF ensures extraction of the tissue-specific pattern theoretically. More theories and performance comparison based on simulated data are needed.

In response to this and comments from Reviewer 2, we have now amended the manuscript to include analyses with simulated data. This includes overall performance in identifying patterns of tissue-specificity and comparisons with other matrix factorization approaches to motivate the specific matrix factorization modeling choices we made in sn-spMF including sparsity and semi-non-negativity.

For simulated data, in addition to sn-spMF, we ran singular value decomposition (SVD) as a baseline model, non-negative matrix factorization (NMF), softImpute (Mazumder et al 2010), sparse SVD (SSVD) (Yang et al 2014), penalized matrix factorization (PMD, with two parameter settings) (Witten et al., 2009), nonparametric Bayesian sparse factor analysis (NBSPA) (Knowles and Ghahramani, 2011), and three versions of flashr methods (Wang et al. 2018) - flashr using the default function (referred to here as flashr_default), flashr with greedily added factors followed by backfitting (referred to here as flashr_bf), and flashr with the non-negative constraint (referred to here as flashr_NN). Details of parameter choices for each method can be found in the updated Methods. We have included the results of this analysis in the main text ("Impact of matrix factorization methodological choices") as follows:

"To evaluate the performance of these methods, we computed the correlation between the learned loadings and the true loadings, and the correlation between the learned factors and the true factors. We observed that sn-spMF and flashr_NN achieve the most accurate loading matrix and factor matrix (Additional file 1: Fig. S15, S16), followed by other flashr approaches, NBSPA and softImpute. Sparsity appears to confer some benefit in accuracy and interpretability of factors."

The theoretical properties of matrix factorization, including its ability to identify non-mutually exclusive groups of features robustly from data, along with methods for tuning hyper-parameters have been well-described in previous work (Brunet et al 2004, Bishop 2006, Wu et al 2016). Sparse priors and their advantages in interpretability and efficiency are similarly well studied (Tibshirani 1996, Efron et al. 2002, Zou et al. 2005, Bishop 2006).

2. A relevant concern on "... matrix factorization allows multiple factors and tissues to be involved in explaining the effect size of an eQTL ...". What are the actual benefits of using multiple factors and tissues to explain the effect size of an eQTL?

We thank the reviewer for this comment. From a biological perspective, in addition to affecting tissues with clear biological similarity that would be captured together as a factor, an eQTL may have an effect in multiple, quite distinct conditions. Transcription factors are often re-used in diverse pathways in various cell types, and variants within these TF binding sites may have eQTL effects in unexpected combinations of conditions. In fact, we do see support for such patterns in eQTLs in the GTEx dataset. To better illustrate the importance of allowing multiple factors for each eQTL, we added examples of eQTLs that exist in multiple, relatively unrelated tissues in Additional file 1: Fig. S4. For example, an eQTL - ENSG00000196730.12 : chr9_87444328 was found in pancreas, stomach, and whole blood from GTEx eQTL analysis (The GTEx Consortium 2020, in submission), a set of tissues that is not similar enough overall to be captured by manual clustering or represented as a factor. This eQTL was indeed identified to have effects in the pancreas-specific factor, stomach-specific factor, and whole blood specific factor from sn-spMF.

As a result of performing matrix factorization, 1,221,109 eQTLs (23% of tested eQTLs) load on at least two tissue-specific factors (Fig. 3d), and 638,784 eQTLs load on both the ubiquitous factor and tissue-specific factors (59% of the u-eQTLs and 22% of the ts-eQTLs, Fig. 3c). This indicates that matrix factorization faithfully captures the feature of eQTLs that an eQTL can affect multiple tissue groups.

3. i). In the Background Section, the authors states that "This method, while easily implemented, requires subjective thresholds and ignores the underlying similarity of tissues." For a statistical inference, it is natural that one needs p-value thresholds.

We regret any lack of clarity in this statement. It is true that p-value thresholds are commonly used in statistical inference, and these are used for sn-spMF as well. However, in order to manually define tissue-specificity in heuristic approaches, one needs more than a standard, single p-value threshold to define significance -- rather, multiple hand chosen thresholds must be specified, as noted in more detail to our response to Reviewer 1, comment 1.i).

ii). It is unclear how the sn-spMF is able to handle similarity of tissues theoretically. The tissues of known biological similarity can be identified by conventional cluster analysis methods.

We agree with the point that tissues or conditions could be clustered rather than factorized. Both clustering and matrix factorization automatically determine patterns of tissue-sharing from data, and are deeply related to each other. However, conventional cluster analysis assigns features to mutually exclusive groups and does not allow a feature (such as a tissue or condition) to appear in more than one cluster. In many biologically relevant scenarios, one feature can be relevant in multiple patterns. For example, in standard gene expression analysis, many genes have more than one function, and should be grouped with more than one set of related genes. For eQTL analysis likewise, a tissue may naturally belong to more than one overlapping group/factor, as

described in point 1-ii)-1) above, and a variant may have effects corresponding to more than one factor as noted in point 2 above. For future use in situations like time-series data, the feature (time) gradually changes, and one time point might look like an intermediate between distinct states or endpoints, and simply clustering time points would be inappropriate. Single cell data will present similar challenges, where a given cell may be affected by a combination of states/conditions along with cell type (Erbe et al. 2020), and is likely to be impacted by genetic effects specific to each of these overlapping factors as well.

4. Instead, there are also factors leading to unstable results by sn-spMF including the choices of dimension K , α and λ .

We regret any lack of clarity on this point. We estimated the parameters in sn-spMF using a fully automated procedure that identifies parameter settings leading to the most stable decomposition as well as the most independent factors (see Methods, Brunet et al 2004, Wu et al 2016). For heuristic approaches, however, there is no objective way to decide the parameters/thresholds.

5. More importantly, this reviewer cannot see new findings and insights by sn-spMF. The relevant results of ts-eQTLs and u-eQTLs are not really novel.

We disagree with the reviewer about the novelty of our results, which include:

- 1) Technical novelty: The sn-spMF method is novel, and made freely available for application to future eQTL studies including studies across cell types, time-series data, single cells data, etc. The method, while based on established theory of matrix factorization, is a tailored version using sparse priors, appropriate non-negativity constraints, and an optimization procedure that allows it to be applied successfully to the eQTL setting. Basic enrichment results follow expected but desirable patterns for tissue-specificity, demonstrating advantages over other approaches and support its future use.
- 2) Biological novelty: Our TF binding site analysis (Fig. 5a) and the resulting lists of TFs provides novel, testable hypotheses regarding the impact of particular transcription factors on tissue-specific genetic regulation of gene expression. We also highlight a subset of example eQTLs where we are able to connect TF binding to tissue-specific effects.

6. I am stumped for the assumption of the model, "the effect of an eQTL across tissues is a linear combination of factors, where every factor represents a common pattern of eQTL sharing across particular sets of tissues ...". What do you mean by "the effect of an eQTL across tissues is a linear combination of factors"?

This is an important point, and we regret any lack of clarity. We have clarified the high-level description in the main text (“Matrix factorization of multi-tissue eQTL effects”) as follows:

“The model overall seeks to decompose an input matrix of eQTL effect sizes in each tissue (regression parameters from a linear model for eQTL mapping) into underlying patterns of tissue-sharing and tissue-specificity. This model assumes that the effect size vector of one eQTL across tissues can be approximated as a linear combination (weighted sum) of learned “factors”, where every factor is a vector representing one common pattern of eQTL effect-sizes across tissues (Fig. 1a). When many entries in the factor are small or zero, as our model will enforce, a factor points to a subset of tissues that are commonly affected by the same eQTLs. Then, for a given eQTL, the loadings, or “weights,” on each factor reflect how strongly that eQTL’s effects are explained by that factor (and corresponding non-zero tissues).”

This describes standard matrix factorization where an input data matrix is decomposed into factors and loadings (revised Figure 1) to capture informative patterns in a lower-dimensional representation. Furthermore, the complete mathematical framework is described in the Methods section (Assignment of eQTLs to factors). For convenience, a brief summary is repeated here:

We denote the input matrix $X \in R^{N \times T}$, where N is the number of eQTLs and T is the number of tissues in the eQTL dataset of interest. Each entry in X corresponds to an eQTL effect size -- that is, the regression coefficient from the linear model association test performed during eQTL mapping. Thus, the effect of an eQTL i across tissues is a row in the X matrix $x_i^T \in R^{1 \times T}$, which matrix factorization approximates as a weighted sum of K learned factors

$f_k^T \in R^{1 \times T}$ ($k = 1 \dots K$), weighted by the learned loadings for this eQTL l_{ik} ($k = 1 \dots K$), such that:

$$x_i^T \approx l_{i1}f_1^T + l_{i2}f_2^T + \dots + l_{iK}f_K^T$$

where f_k is the k^{th} factor (that is the k^{th} column in the factor matrix $F \in R^{T \times K}$), l_{ik} is the k^{th} element in the weights l_i (that is the i^{th} row in the loading matrix $L \in R^{N \times K}$). The above describes the general framework of standard matrix factorization, which sn-spMF then modifies with a non-negativity constraint on F and sparsity-inducing priors on both F and L (We have clarified the constraints of sn-spMF in the main text (“Methods - Weighted sparse semi-nonnegative matrix factorization algorithm”))

7. i). Conceptually, being tissue specific is a relative measurement. It is unclear how the degree of the eQTL tissue specificity can be quantified by sn-spMF.

We thank the reviewer for this comment. As described in “Methods - Assignment of eQTLs to factors”, sn-spMF assesses tissue-specificity by determining whether a tissue-specific factor is statistically needed to explain the eQTL effect sizes of that SNP across tissues, assessing statistical significance in a multivariate regression. The “degree” of tissue-specificity, or importance of a given factor, can be evaluated by the magnitude of the corresponding

regression coefficient, but a binary decision is also made based on statistical significance (as commonly employed in association tests including the heuristic methods). For each eQTL, sn-spMF therefore provides a) magnitude of contribution for each factor b) statistical significance for each factor and c) the total number of factors used for the eQTL, each of which may be of interest to a researcher seeking to understand the tissue-specificity of a given eQTL.

ii). In the results section, the authors stated that sn-spMF detected more biologically coherent eQTLs than heuristic thresholding method. It is unclear how the authors defined a tissue specific eQTLs by the heuristic thresholding method. For example, if the eQTL is specific for multiple brain tissues only, it could be still called as being tissue specific. If the authors require it can be only specific for a single brain tissue, many tissue specific eQTLs may be missed.

We appreciate the importance of this question, so in response to this comment and comment 1 from Reviewer 2, we have added a comparison to a second heuristic approach where we attempt to manually define subsets of tissues (heuristic_2, see Methods). Tissues with a clear shared biology have been grouped together, such as the group of brain tissues noted by the reviewer (Additional file 2: Table S1). One complication this introduces is that beyond the four thresholds already needed for heuristic_1, one additional threshold must be chosen manually with no clear guidance, specifically the minimum number or fraction of “in-group” tissues of interest for which the eQTL must appear in order to be counted as a hit for this group. Here, we set this threshold to be 50% of the tissues in a subgroup. We observed that there were indeed a subset of eQTLs captured as ts-eQTLs by heuristic_2, but not heuristic_1, especially for groups with more tissues such as brain tissues. Two such eQTLs were shown in Additional file 1: Fig. S6. However, because of the additional threshold, there were *overall* fewer brain-specific eQTLs identified in heuristic_2 compared to the union of ts-eQTLs across brain regions for heuristic_1. The same downstream analysis was run for both heuristics. Heuristic_2 shows some modest benefits over heuristic_1 particularly for tissue-specific enhancer enrichment and TFBS identification (Additional file 2: Table S2, S4, S5, S6, S7). The number of ts-eQTLs identified by both heuristic methods remains much lower than for sn-spMF (Additional file 1: Fig. S3, S5). sn-spMF performs comparably or better on enrichment metrics, even with its much larger set of ts-eQTLs. Finally, sn-spMF and other matrix factorization methods identify far more candidate TFBS underlying ts-eQTLs than either heuristic approach (Additional file 2: Table S7). Matrix factorization approaches that automatically learn overlapping patterns of tissue-sharing continue to offer an advantage.

Also, heuristic methods would be hard to implement in situations where we have limited a priori knowledge about the relationship between conditions (in contrast to our knowledge of tissue similarity). For example, in time-series data, it's impossible for us to know how patterns change over time. Another example is in single cell data, where cell states and cell types are often not known ahead of time.

8. It is unclear how the statistical significance of tissue-specific patterns can be evaluated by sn-spMF. Is it done here "...Statistical importance is captured by the P values of the factors...."? How the P-values are calculated?

We have clarified the wording of this sentence in the main text ("Methods - Assignment of eQTLs to factors") as follows:

"For each eQTL, the weighted linear regression is fit: $x = Fl$ weighted by its reciprocal of standard error. Statistical significance of each factor for the eQTL is determined according to P values based on the standard t-test from this linear regression."

9. How robust is the method to weak LD?

We thank the reviewer for raising this important point. The method itself doesn't deal with weak LD, but depends on an appropriate input SNP set provided by the user. Here, we restricted the analysis of ts-eQTLs and u-eQTLs to eQTLs within "credible sets" of SNPs likely to contain causal eQTL variants, obtained by applying published fine-mapping methods (Wakefield, 2009, see Methods) to the GTEx eQTL statistics. eQTLs' associations resulting from weak LD with a causal variant are thus less likely to enter the analysis. Also, to obtain the factor matrix, we used the top SNP per gene from the credible sets as input (see Methods). We recommend users of our method similarly restrict to credible sets or other strategies to avoid impact of weak LD on the results.

10. What types of eQTL association statistics in the matrix? Please state clearly.

We thank the reviewer for this comment. Input matrix X contains the effect sizes published by the GTEx Consortium, which are the regression coefficients from standard linear model association testing between the SNP genotype and gene expression levels in each tissue. Matrix W contains the reciprocal of standard error on this same regression coefficient.

11. It may be interesting to know tissue selective expression affect the tissue-selective eQTL as the eQTL of genes with tissue specific expression has been applied to estimate disease-relevant tissues previously (Nat Genet 2017, 49:1676-1683 and Nat Genet 2018, 50:621-629).

We thank the reviewer for this suggestion. It is a very interesting question, and we added references to two relevant publications (see Background). We have not explicitly analyzed the effect of tissue-specific gene expression here, per se, but rather tissue-specific genetic effects on expression. Note that in the GTEx eQTL statistics, eQTLs from genes with very low expression in particular tissues correspond to "missing" values for those tissues, because GTEx simply did not assess eQTLs for lowly expressed genes. Missing values in the input matrix X do not impact the sn-spMF objective function. Therefore, genes with very low expression in

particular tissues will not impact our factors or loadings. However, it is a valuable point that could perhaps be exploited in a future extension of our method or application to disease.

Reviewer #2: The authors developed a matrix factorization model and applied to GTEx dataset to capture patterns of eQTLs across different tissues. The method is a constrained weighted semi-nonnegative sparse matrix factorization, named sn-spMF.

The results given by the factorization of 49 tissues resulted in 23 factors, many clearly representing specific tissues, others representing a combination of some similar tissues and one factor shared among all tissues (called universal).

Thank you for nicely summarizing our method.

1. The comparison of sn-spMF with other methods is limited to two other approaches. The authors start by comparing their method of classifying eQTLs into factors with simple thresholding heuristics. Clear advantages are shown for using matrix factors in terms of gene enrichment relevance and regulatory element enrichments. It's clear that by using matrix factorization the authors managed to derive an unbiased way of aggregating similar tissues and clearly distinguish tissue-specific eQTLs. However, the comparison with heuristic methods could be performed in an opposite way, by carefully selecting relevant tissues and thresholds in a "gold-standard" fashion and then comparing it with the unbiased factorization to measure how reproducible the results are.

We thank the reviewer for this comment. Manually defining tissue groupings and appropriate thresholds has proven quite challenging to do well for several reasons. Identifying tissue-specific eQTLs using heuristic methods involves deciding on a series of heuristics and thresholds including A) manual groupings of tissues/conditions that are expected to share eQTLs a priori, B) lower bound cutoff for significance in the tissue(s) of interest, C) lower bound on the number/fraction of tissue(s) of interest for which the eQTL must appear D) upper bounds on the effect size observed in "irrelevant" tissues and E) upper bounds on the number of other "irrelevant" tissues for which the eQTL may appear, all needed to decide if the eQTL is truly specific to the tissue(s) of interest. There is no clear automatic approach for making these choices.

However, in response to this important comment and a related concern from Reviewer 1 (point 7.ii), we have added a comparison to a second heuristic approach using manually defined subsets of tissues (heuristic_2, see Methods). Tissues with clear shared biology, such as the brain regions, have been grouped together (Additional file 2: Table S3), and the same downstream analysis was run. While several example eQTLs corresponding to these tissue groups were detected (Additional file 1: Fig. S6), overall, no major increase in biologically meaningful enrichment was observed over heuristic_1 (Additional file 2: Table S2, S4, S5, S6, S7). This appears to be due to difficulty in setting thresholds C and E optimally, where many

interesting eQTL are excluded, and there isn't an overall increase in the number of ts-eQTLs identified.

Please see Table S2, S4, S5, S6, S7 within this response for a convenient summary comparing between all methods.

Also, heuristic methods would be hard to implement in situations where we have limited a priori knowledge about the relationship between conditions (in contrast to our knowledge of tissue similarity). For example, in time-series data, it's impossible for us to know how patterns change over time. Another example is in single cell data, where cell states and cell types are often not known ahead of time.

2. Next, the authors compared the sn-spMF with flashr (an Empirical Bayes matrix factorization method). According to them, by not using nonnegativity constraints, flashr factors were not so clear to distinguish tissues. How flashr was run is not described. It's possible to run flashr with nonnegativity constraints as well, maybe it would be good to compare how it performs as well. Also, it's not clear how the model was fitted. The authors run the greedy algorithm or they applied backfitting to improve the fit?

We regret this lack of clarity in the original manuscript. We originally ran flashr using the default function (*flashr*, referred to here as *flashr_default*). In response to this comment, we also tried the version with greedily added factors followed by backfitting (using functions: *flash_greedy_workhorse* and *flash_backfit_workhorse*, referred to here as *flashr_bf*), and flashr with the non-negative constraint (referred to here as *flashr_NN*). We communicated with the authors of flashr to make sure these were run correctly and with the most recent code.

We found that *flashr_bf* and *flashr_NN* result in increased sparsity compared to *flashr_default*, but still more dense than sn-spMF, with more tissues appearing in each factor (Additional file 1: Fig. S18, 19). Overall, the results of *flashr_bf* do offer an improvement over *flashr_default*, including a better fit to the eQTL effect sizes (Additional file 1: Fig. S27; Additional file 2: Table S4), and better enrichment for regulatory elements, genes, and TFBS (Additional file 2: Table S5, S6, S7). On the other hand, *flashr_NN* resulted in degraded performance compared to *flashr_default* (Additional file 1: Fig. S17, S20, S23, S24; Additional file 2: Table S5, S6, S7). Thus, we have edited the manuscript to focus on *flashr_bf* results as the primary comparison with sn-spMF, though all results for all three methods are included in the supplement.

sn-spMF outperforms all versions of flashr on TFBS enrichment and enrichment of ts-eQTLs in tissue-specific enhancer elements (better distinguishing them from u-eQTLs), whereas flashr performs well on GO enrichment, broadly consistent with the findings in the original manuscript, but now confirmed for multiple variations on flashr. We have updated the text to reflect the improved performance of *flashr_bf* and included all results in the Supplement (Additional file 1: Fig. S17, S20, S23, S24; Additional file 2: Table S5, S6, S7).

Investigating multiple versions of flashr did reveal some interesting patterns worth highlighting. The increased sparsity of flashr_bf did appear to offer improvements over the flashr_default, as we suspected benefitted sn-spMF. However, flashr_bf does not include non-negativity constraints on the factors, thus complicating interpretation of latent patterns and tissue specificity. For example, we found that factors that contain tissues with different signs do not correspond well to patterns in the actual eQTL effect sizes - only 19% - 35% of eQTLs that mapped to such mixed sign factors actually display opposite sign effects in the predicted tissues (Additional file 1: Fig. S21). The degraded performance of flashr_bf compared to sn-spMF on TFBS and regulatory enrichment likely arises from the lack of sparsity and the poor isolation of tissues that are jointly affected. Thus, flashr_bf factors cannot be directly interrogated to understand common patterns across tissues or tissue-specificity of individual eQTLs. flashr_NN on the other hand produced sparse, highly interpretable tissue factors, but did a poor job of fitting the actual eQTL effect sizes compared to flashr_default, flashr_bf or sn-spMF (Additional file 1: Fig S27). We also observed high multicollinearity of the flashr_NN ubiquitous factor with the tissue-specific factors, which made it difficult to distinguish tissue-specific effects across numerous factors from truly ubiquitous effects. This is indicated by *variance inflation factor* of the ubiquitous factor $VIF = \frac{1}{1-R^2}$, where R^2 denotes the coefficient of determination obtained by fitting a regression model for the ubiquitous factor on all other factors. We observed a very high VIF of 46 for flashr_NN, compared to 1.5 for flashr_default and 4.5 for sn-spMF, where guidelines suggest that a *VIF* higher than 10 may lead to imprecise estimates of the regression coefficients (Yoo et al. 2014, Stine 1998).

We have added descriptions of these observations, and have added a note in the section “Discussions and Conclusions” indicating that flashr does have the advantage of automatically learning the parameters with less computational burden, compared to sn-spMF where grid search is needed for tuning parameters, as follows:

“Other versions of matrix factorization, such as flashr, also provide meaningful views of tissue specificity. In particular, we note the flashr has the advantage of learning the parameters with less computational burden, compared to sn-spMF where a grid search is needed for tuning parameters.”

Please see Table S2, S4, S5, S6, S7 within this response for a convenient summary comparing between all methods.

3. There is no other method that could be used for comparison? For example, in flashr manuscript, they compared their method with Sparse Factor Analysis (SFA) (Engelhardt and Stephens, 2010), SFAmix (Gao et al., 2013), Nonparametric Bayesian Sparse Factor Analysis (NBSFA) (Knowles and Ghahramani, 2011), Penalized Matrix Decomposition (Witten et al., 2009) (PMD, implemented in the R package *PMA*), and Sparse SVD (Yang et al., 2014) (SSVD, implemented in R package *ssvd*).

In response to this important comment, we have now compared our method to several matrix factorization methods.

For simulated data, in addition to sn-spMF and the three flashr methods discussed above (Wang et al. 2018), we ran singular value decomposition (SVD) as a baseline model, non-negative matrix factorization (NMF), softImpute (Mazumder et al 2010), SSVD (Yang et al 2014), penalized matrix factorization (PMD, with two parameter settings) (Witten et al., 2009), and nonparametric Bayesian sparse factor analysis (NBSPA) (Knowles and Ghahramani, 2011) as the reviewer suggested. Details of parameter choices for each method can be found in the updated Methods. We have included the results of this analysis in the main text (“Impact of matrix factorization methodological choices”) as follows:

“To evaluate the performance of these methods, we computed the correlation between the learned loadings and the true loadings, and the correlation between the learned factors and the true factors. We observed that sn-spMF and flashr_NN achieve the most accurate loading matrix and factor matrix (Additional file 1: Fig. S15, S16), followed by other flashr approaches, NBSPA and softImpute. Sparsity appears to confer some benefit in accuracy and interpretability of factors.”

Beyond simulation, we also added additional comparisons using the GTEx data. However, not all of the methods used for simulation can take input data that includes missing values. A large fraction of GTEx eQTLs are only tested in a subset of tissues, and removing eQTLs with missing data would lead to too few eQTLs available for downstream enrichment analysis for real data. Therefore, in this analysis, we compare only flashr methods, sn-spMF, softImpute, and PMD, each of which can handle missing data. Flashr results are discussed in detail in the previous point. We observed that factors learned from softImpute and PMD are more dense than factors learned by sn-spMF, and are thus hard to interpret regarding tissue-specificity (Additional file 1: Fig. S24, S25, S26). Downstream analysis using results from softImpute and PMD demonstrated:

- 1). For softImpute and PMD, u-eQTLs and ts-eQTLs are not well distinguished from each other according to cis-regulatory element enrichment, making tissue-specificity difficult to interpret (Additional file 1: Fig. S28; Additional file 2: Table S6). Importantly, cis-regulatory regions for matched tissues are not more enriched for ts-eQTLs than for un-matched tissues (Additional file 1: Fig. S28). softImpute and PMD thus result in poor distinction between u-eQTLs and ts-eQTLs, which we hypothesize is due to the lack of sparsity in the factors not well-capturing tissue-specific effects.

- 2). The number of GO pathways enriched for the ts-eGenes is comparable to flashr, and a bit more than sn-spMF. This may be due to power, where a greater number of genes are present in the background sets for this test when a large number of tissues are included in a factor.

3) Compared to sn-spMF, similar numbers of TFs are enriched in promoter regions (Additional file 2: Table S6, S7). Critically, however, fewer TFs are enriched in enhancer regions for ts-eQTLs for PMD and softImpute compared to sn-spMF, indicating again the possibility that the ts-eQTLs are simply not highly tissue-specific. Also, it is more challenging to interpret tissue-specific regulatory mechanisms even for the enriched TFs that do come up for these methods, because each factor contains so many tissues.

Please see Table S2, S4, S5, S6, S7 within this response for a convenient summary comparing between all methods.

4. And how about other NMF models?

This is a good question. NMF models require all input values to be non-negative. This assumption doesn't apply naturally to eQTL effect sizes, which do have varying directions of effect in different tissues. The simulations performed above demonstrated poor performance for NMF in such cases, where we took the absolute value of the input matrix in order to meet the constraints of NMF. For real eQTL data, taking the absolute value of effect sizes spuriously increases apparent similarity between tissues and increases the possibility of false detection of u-eQTLs. Other sparse matrix factorization methods including flashr offer better performance.

Few commentaries about the figures.

All figure legends can be improved.

We thank the reviewer for the comment. We proofread and improved the figure legends.

Figure 1A is not very clear or informative.

In response to this comment, we have modified Figure 1A to indicate simplified examples of eQTLs with additive effects from latent patterns.

In Figure 2 the loading bar at the right side of the table could be aligned with the x-axis of that table to match respective factors. Also, fading colors to show no significant results is not good to see. Maybe using a "*" in the significant cells can improve the visualization.

We appreciate the reviewer's concern. We think that aligning the loading bar with the x-axis of the factor matrix would make the specific operation (matrix multiplication) less intuitive.

Following the reviewer's advice, we have combined fading colors with "*" to help to make the significant results more obvious.

Other minor points:

I understand the concept of Universal for the eQTLs shared across tissues. However, the authors are using 49 tissues, and they are not all tissues from the human body. The word ubiquitous defined as "present or found in all tissues in the study" can be more appropriate.

We now use the word ubiquitous throughout the text.

Page 3, second paragraph: "... 8 tissues are not significantly represented by any tissue-specific factor and, therefore, can't be captured in this analysis." Which tissues? How many samples by each of those tissues?

We thank the reviewer for pointing this out. We have added a supplementary table to list the 8 tissues that are not captured along with their sample size (Additional file 2: Table S1). Compared to the other 41 tissues, these tissues do have significantly fewer samples (t-test P-value < 0.024).

In the methods, the section of the factorization algorithm can be better explained. For example, it's not clear what is the D in the formula. Also, the authors should include the W matrix in Figure 1 of the paper.

We apologize for this lack of clarity. We have now included the weight matrix in Figure 1, and added a sentence to explain D, as in the legend ("*D is the number of eQTLs*").

Fig 6C, should include a p-value to show significance.

We thank the reviewer for pointing this out. We included the p-value from GTEx eQTL analysis in the plot.

Supplementary Table 2. Number of u-eQTLs and ts-eQTLs captured by different methods

	u-eQTLs	ts-eQTLs
sn_spMF	1,076,761	76,976 - 431,585
heuristic_1	312,502	1,374 - 102,414
heuristic_2	175,637	1,460 - 201,584
flashr_backfitting	1,929,939	69,594 - 929,009

flashr_default	1,785,127	55,295 - 701,035
flashr_NN	243,467	54,306 - 338,615
softImpute	1,936,985	62,012 - 1,006,031
PMD_cv1	1,937,676	56,857 - 986,481
PMD_cv2	1,945,235	84,409 - 1,181,099

Supplementary Table 4. Proportion of all tested eQTLs that have R^2 between model-predicted and actual effect sizes above a specific threshold:

	$R^2 > 0$	$R^2 > 0.2$	$R^2 > 0.6$
sn_spMF	59%	50%	21%
flashr_backfitting	62%	51%	22%
flashr_default	57%	47%	20%
flashr_NN	43%	34%	10%
softImpute	55%	44%	20%
PMD_cv1	55%	42%	19%
PMD_cv2	58%	45%	20%

Supplementary Table 5. Enrichment of u-eQTLs and ts-eQTLs in cis-regulatory regions

	OR in promoter		OR in enhancer	
	u-eQTLs	ts-eQTLs	u-eQTLs	ts-eQTLs
sn_spMF	1.9	1.5	1.0	1.3
heuristic_1	3.0	1.2	1.1	1.3
heuristic_2	3.1	1.2	1.1	1.4
flashr_backfitting	1.8	1.6	1.1	1.1

flashr_default	1.8	1.6	1.1	1.1
flashr_NN	2.1	1.5	0.9	1.1
softImpute	1.8	1.7	1.1	1.1
PMD_cv1	1.8	1.8	1.1	1.1
PMD_cv2	1.8	1.7	1.1	1.0

Supplementary Table 6. Number of enriched GO pathways for ts-eGenes

	All ts-eGenes	Strictly defined ts-eGenes
sn_spMF	546	45
heuristic_1	110	0
heuristic_2	421	1
flashr_backfitting	593	101
flashr_default	642	93
flashr_NN	453	7
softImpute	659	90
PMD_cv1	615	84
PMD_cv2	556	102

Supplementary Table 7. Number of enriched TFBS for u-eQTLs and ts-eQTLs in cis-regulatory regions

	Promoter		Enhancer	
	u-eQTLs	ts-eQTLs	u-eQTLs	ts-eQTLs
sn_spMF	136	181	39	264
heuristic_1	59	5	8	47
heuristic_2	97	9	4	54
flashr_backfitting	143	178	99	165

flashr_default	136	137	90	123
flashr_NN	70	104	2	109
softImpute	157	196	113	169
PMD_cv1	147	191	107	166
PMD_cv2	160	203	111	167

Reference:

Bishop, Christopher M. Pattern Recognition and Machine Learning. New York :Springer, 2006.

Brunet, J.P., Tamayo, P., R Golub, T., P Mesirov, J.: Metagenes and molecular pattern discovery using matrix factorization. Proceedings of the National Academy of Sciences 101, 4164–9 (2004)

Efron, B., Hastie, T., Johnstone, L., Tibshirani, R.: Least angle regression. The Annals of Statistics 32 (2002).

Erbe, R., Kessler, M., Favorov, A., Easwaran, H., Gaykalova, D., Fertig, E.: Matrix Factorization and Transfer Learning Uncover Regulatory Biology Across Multiple Single-cell ATAC-seq Data Sets.

Hastie, T., Mazumder, R.: softimpute: Matrix completion via iterative soft-thresholded svd (2015). R package version 1.4

Knowles, D., Ghahramani, Z.: Nonparametric bayesian sparse factor models with application to gene expression modeling. Annals of Applied Statistics - ANN APPL STAT 5 (2010).

Mazumder, R., Hastie, T., Tibshirani, R.: Spectral regularization algorithms for learning large incomplete matrices. Journal of machine learning research : JMLR 11, 2287–2322 (2010)

Stephens, M., Wang, W., Willwerscheid, J.: flashr: Empirical bayes matrix factorization (2019). R package version 0.6-7

Stine, R.: Graphical interpretation of variance inflation factors. The American Statistician, 53–56 (1995)

Tibshirani, R.: Regression shrinkage and selection via the lasso. *Journal of the Royal Statistical Society Series B* 58, 267–288 (1996).

Wang, W., Stephens, M.: Empirical bayes matrix factorization. arXiv:1802.06931 (2018)

Witten, D., Tibshirani, R., Hastie, T.: A penalized matrix decomposition, with applications to sparse principal components and canonical correlation analysis. *Biostatistics (Oxford, England)* 10,, 515–34 (2009).

Witten, D., Tibshirani, R., Gross, S., Narasimhan, B.: Pma: Penalized multivariate analysis (2019). R package version 1.1

Wu, S., Joseph, A., S. Hammonds, A., E. Celniker, S., Yu, B., Frise, E.: Stability-driven nonnegative matrix factorization to interpret spatial gene expression and build local gene networks. *Proceedings of the National Academy of Sciences* 113, 201521171 (2016)

Yang, D., Ma, Z., Buja, A.: A sparse singular value decomposition method for high-dimensional data. *Journal of Computational and Graphical Statistics* 23 (2014).

Yang, D.: ssvd: Sparse svd (2013). R package version 1.0

Yoo, W., Mayberry, R., Bae, S., Singh, K., He, Q., Lillard, J.: A study of the effects of multicollinearity in multivariable analysis. *International journal of applied science and technology* 4, 9–19 (2014)

Zou, H., Hastie, T.: Regularization and variable selection via the elastic net. *Journal of the Royal Statistical Society Series B* 67, 768–768 (2005).

Second round of review

Reviewer #1: This reviewer appreciates the authors for the detailed responses. Multiple my previous comments have been addressed. However, there remain some unclear.

1. The first response makes this reviewer worry about the physical definition of "ubiquitous eQTLs" (u-eQTLs) and "tissue-specific eQTLs" (ts-eQTLs) in the manuscript as the ts-hidden factors looks abstract. For example, how many tissues should an eQTLs occur among the 49 human tissues at least when it is called a u-eQTLs? How many tissues should an eQTLs occur among the 49 human tissues at most when it is called a ts-eQTLs?

We thank the reviewer for this comment. In the first response, we addressed the difficulty of defining thresholds to call ts-eQTLs and u-eQTLs using heuristic methods. The matrix factorization method, however, is able to automatically learn a universal factor shared across tissues and tissue-specific factors from the data and map eQTLs to the factors; it doesn't require a predetermined number of tissues to define u-eQTLs and ts-eQTLs. There is, in the learned factor matrix, a single clear universal factor (Fig. 1b, Fig. 2, Additional file 1: Fig. S2). All other factors, we term tissue-specific - while in theory it could be the case that the other factors involve many tissues and it would be hard to draw a line, in practice, our sparse factors involve no more than 13 tissues and are thus clearly tissue-specific. As we pointed out in the manuscript: *the eQTLs that load on the ubiquitous factor are referred as "ubiquitous eQTLs", and the eQTLs that have significant loadings on each of the tissue-specific factor are referred as "tissue-specific eQTLs" (ts-eQTLs)*. Fig. 2 illustrates how the ubiquitous effect and tissue-specific effect are captured.

2. "The number of ts-eQTLs identified by both heuristic methods remains much lower than for sn-spMF (Additional file 1: Fig. S3, S5). sn-spMF performs comparably or better on enrichment metrics, even with its much larger set of ts-eQTLs. Finally, sn-spMF and other matrix factorization methods identify far more candidate TFBS underlying ts-eQTLs than either heuristic approach (Additional file 2: Table S7)" The performance comparison seems too complex unnecessarily. The author only need show that the proposed method can more accurately detect the assumed true ts-eQTL and u-eQTL respectively in simulations.

Following the reviewer's useful suggestion, we mapped the eQTLs to learned factors on simulated data and computed the recall and precision for u-eQTLs and ts-eQTLs (Additional file 1: Fig. S18). We observe that sn-spMF and flashr_NN achieve the highest precision and recall for identifying u-eQTLs and ts-eQTLs, which aligns with our previous observation that these two models achieve the most accurate loading matrix and factor matrix. Code and simulated data used to generate the evaluation statistics are available in the Github repository, under the folder "simulation".

While we agree that simulations provide straightforward comparisons, we think application to real data provides additional support to demonstrate the tool's utility. Simulations may not capture properties of real eQTL data accurately, since real effect sizes and tissue sharing are unknown. Indeed, one of the main advantages of sn-spMF is that it provides patterns with better biological interpretability and results in ts-eQTLs and u-eQTLs with higher functional enrichments.

3. A relevant problem is that the enrichment of ts-eQTL in cis-regulatory regions TF binding sites and GO biological processes seems new findings, which may be not suitable to be used as benchmark for method comparison.

We appreciate the reviewer's comment. While TF binding and GO processes are new findings, they also support the performance of sn_spMF on a real application, and do have quantitative

assessments of enrichment that can be used for comparison. We have therefore opted to include these in our evaluation. Simulations are included and expanded in order to show comparisons using a known gold standard, but real applications better reflect the true properties of data that future users would encounter, so both are important for potential users to see when evaluating the method.

4. This reviewer still think it is important investigate whether tissue specific expression gene tends to have ts-eQTL after excluding the lowly expressed genes. It is also important to SYSTEMATICALLY investigate what factors contribute to the tissue specific genetic effects on expression. What are the biological principles?

We thank the reviewer for raising this important question. We agree that it is a very interesting question to systematically explore the biological principles for tissue-specific eQTLs. Indeed, we had previously explored their biological principles in three ways in the manuscript:

1). Ts-eQTL genes are enriched for GO terms, including GO terms most relevant to the corresponding tissues (Fig. 4; Additional file 1: Fig. S9, S10, S11). Thus genes participating in tissue-specific biological pathways are more likely to have ts-eQTLs.

2). Ts-eQTLs are enriched in enhancer regions of their corresponding tissues, while u-eQTLs are enriched in promoter regions (Additional file 1: Fig. S13, S14) (S14 is added in the revision following the reviewer's suggestion).

3). We also identified transcription factor binding sites enriched for the ts-eQTLs, which may provide insight about tissue-specific regulation. This suggests that SNPs that disrupt the binding sites for tissue-specific TFs can lead to tissue-specific eQTLs.

Given the importance of this reviewer's point, we performed the following additional analyses in order to explore factors contributing to tissue specificity, and to demonstrate the effect of tissue-specific gene expression on ts-eQTLs:

First, we followed the reviewer's suggestion to explore the effect of removing genes with low expression on ts-eQTLs, and showed the proportion of genes with ts-eQTLs among tested genes after excluding genes with low expression based on different thresholds. In the boxplots, each dot represents the proportion of genes with ts-eQTLs in one tissue after excluding genes with median TPM lower than the threshold in that same tissue. Threshold = 0 means no threshold was used. Excluding genes with median expression below 0.1 rather than 0 results in a noticeable increase of the proportion of genes with ts-eQTLs. However, excluding additional genes with expression below other, higher thresholds results in only a marginal increase in the proportion of genes with ts-eQTLs. This indicates genes with moderately low expression do not exhibit a significant difference in the likelihood of being tissue-specific. This plot is added as Additional file 1: Fig S3.

Finally, we examined the number of tissues whose enhancers and promoters overlap ts-eQTLs and u-eQTLs and added a figure in Additional file 1: Fig. S14. In the boxplot below, the normalized proportion of ts-eQTLs in each bin is graphed on the y-axis, while the number of tissues with an enhancer/promoter overlap is graphed on the x-axis. The same analysis is done for random SNPs matched for u-eQTLs and ts-eQTLs, see Additional file 1: Fig S14 for details. Unlike random SNPs, ts-eQTLs are less likely to overlap promoters shared across many tissues than u-eQTLs. Furthermore, ts-eQTLs are more likely to overlap enhancers found in a small number of tissues than are u-eQTLs (left bottom panel, x-axis 1-8), which is not the case in random SNPs, where the medians hover closer to 1 for the lower numbers of tissues.

5. "the weighted linear regression is fit: $x = FL$ weighted by its reciprocal of standard error". Please explain the rationale of this regression clearly. How is it weighted by its reciprocal of standard error? What x is?

We regret any lack of clarity in this statement. We have clarified the description in the Methods section ("Assignment of eQTLs to factors") as follows:

"After we have learned the factors, we identify a set of relevant factors for each eQTL using weighted linear regression. Specifically, for each eQTL, a weighted linear regression of the form $x = FL$ is fit, where x is the vector of eQTL effect sizes across tissues, F is the factors learned from sn-sPMF, and L are regression coefficients. Weights w are incorporated, where w_t is the reciprocal of the standard error for the eQTL effect size x_t in tissue t . Weighted linear regression using standard error in this manner is a common approach allowing data points with high uncertainty to have less influence on the regression parameter estimates¹."

For reviewer convenience, the details of the weighted linear regression are shown here:

For each eQTL, a weighted linear regression is fit to learn the coefficients l . Let K be the number of factors, T the number of tissues. Suppose x is the vector containing effect sizes of the eQTL across tissues $x \in R^{1 \times T}$, and the t^{th} element in x (denoted by x_t) represents the effect size of the eQTL in tissue t . $w \in R^{1 \times T}$ is the vector of the reciprocals of the standard errors from eQTL analysis (standard error represents uncertainty about the regression parameters from a linear model for eQTL mapping), and the t^{th} element in w (denoted by w_t) represents the standard error of the eQTL in tissue t . $F \in R^{T \times K}$ is the factor matrix learned from sn-spMF, and F_t represents the t^{th} row in F . When fitting the regression $x = F \cdot l$, the loss is weighted by w , such that:

$$loss = \sum_{t=1}^{t=T} (x_t - F_t \cdot l) \times w_t, w_t = \frac{1}{\text{standard error in tissue } t}$$

6. The R scripts (https://github.com/heyuan7676/ts_eQTLs) are not user-friendly. Example data and more documents should be provided. This reviewer does not see the codes for model selection.

We thank the reviewer for raising this important point. Following the reviewer's suggestion, we added detailed documentation on Github (https://github.com/heyuan7676/ts_eQTLs) which includes instructions for running the sn-spMF model (with an optional step of randomized initialization), performing model selection, and mapping eQTLs to the loadings. We demonstrate the process of performing model selection and mapping eQTLs using a subset of eQTLs from GTEx v8. Example input and output files are provided in the repository (data is under the folder "data", and output files are under the folder "output"). In addition, we have added a reproducible demonstration using simulated data in the "simulation/" folder within the Github repository. This directory also contains code and results for performance comparison between sn-spMF and other MF methods on simulated data.

7. By the way, the manuscript looks a methodological paper. However, the title does not look like that.

Thank you for the suggestion, we have converted the manuscript to a Method paper to fit the topic, and have revised the title to *"Matrix factorization informs tissue-specific genetic regulation of gene expression."*

Reviewer #3: In this manuscript, He et al present a novel matrix factorisation method for jointly modelling eQTL summary statistics from multiple cell types, tissues and/or biological contexts

(such as stimulations). The two novel aspects of the model are well captured in its name - semi-nonnegative sparse matrix factorization (sn-spMF). The sparsity constraint of the model forces most factors to have near-zero weights in most cell types or tissues. The semi-nonnegative aspect of the model forces all of the weights of the factors to be non-negative. Both of these aspects help make the modelling results more interpretable by identifying small groups of similar cell types or tissues in which the eQTL effects are most likely to be shared. The non-negative aspect of the model further aligns well with emerging biological understanding that a single causal eQTL variant is unlikely to have opposing effects sizes in different contexts and if observed marginal SNP effects do have opposing effects in different cell types or tissues, this likely due to multiple independent linked causal variants with opposing effects. The authors then demonstrate how the model can be used to distinguish between tissue-specific and shared eQTLs in the GTEx dataset and how these groups of eQTLs can be linked to specific transcription factors. Finally, the authors make their software available as a tool that other researchers can apply to their own datasets.

Since this manuscript was posted on biorxiv, we have already applied it to our internal eQTL dataset covering 12 cell types, tissues and/or conditions. We can confirm that it works well on our dataset, identifying meaningful factors that helps us to distinguish between shared and cell-type-specific eQTLs. We expect it to be similarly useful for ongoing efforts to characterise cell type and context-specific eQTLs using single cell approaches.

General comments

Both initial reviewers raised concerns about the value of matrix factorisation for distinguishing broadly shared eQTLs from those that are specific to a subset of cell types or tissues. However, here I agree with authors that defining such sample groupings a priori can be challenging. It is likely that eQTL sharing between tissues is partially driven by sharing of cell types and cell states between tissues. For example, Glastonbury et al, AJHG 2019 identified eQTLs specific to stimulated monocytes in adipose tissue which are likely to correspond to infiltrating inflammatory macrophages present in the adipose tissue. Similarly, different cell types can share signalling pathways that respond to shared stimuli. For example, we have observed that activation of the NF- κ B signalling in both LCLs and monocytes can lead to the appearance of shared eQTLs in these two cell types that are missing in naïve B-cells or monocytes. This type of sharing can be very tricky to define a priori (unless we have a very good understanding of the underlying biology) and can only be discovered with matrix factorisation or related clustering approaches (as the authors have correctly pointed out in response to the other two reviews). In fact, discovering the underlying biology responsible for such shared eQTL effects (and identifying the upstream transcription factors mediating these effects) is arguably one of the main goals of many large-scale eQTL studies.

I see that the manuscript has already greatly improved in the first round of review. I particularly like the re-designed Figure 1 that explains really well how factors contribute to eQTLs. I believe that this method is going to be clearly valuable for the eQTL community and I believe it will be

widely used by up-coming cell type and context-specific eQTL studies (e.g. the single cell eQTLGen Consortium (<https://elifesciences.org/articles/52155>) and many others).

In general, I am satisfied the authors response to concerns from Reviewers 1 and 2. I have few additional concerns stemming from our experience with using the sn-spMF software.

Major concerns

1. In response to point 5 from Reviewer 1, the authors stress that the main novelty of the manuscript is the new sn-spMF method and I completely agree with this assessment. However, I feel that the utility of the method is strongly hampered by the lack of clear documentation on how to use it. I would strongly recommend you to add a worked out example to the accompanying GitHub repository (https://github.com/heyuan7676/ts_eQTLs) covering all of the necessary steps of your analysis (preparing input files, handling missing values, performing parameter selection for matrix factorisation, running the factorisation and finally assigning eQTLs to factors). You could do this using a (very small) subset of the GTEx data that you have used in the paper that still produces more-or-less meaningful results. Alternatively, you could do this using simulated data if you think that would work better. I completely agree that research code often cannot meet the same standards that are set for production-ready software packages, but I believe that clear documentation is still a must.

We thank the reviewer for raising this important point. Following the reviewer's suggestion, we added detailed documentation on Github (https://github.com/heyuan7676/ts_eQTLs) which includes instructions for running the sn-spMF model (with an optional step of randomized initialization), performing model selection, and mapping eQTLs to the loadings. We demonstrate the process of performing model selection and mapping eQTLs using a subset of eQTLs from GTEx v8. Example input and output files are provided in the repository (data is under the folder "data", and output files are under the folder "output"). In addition, we have added a reproducible demonstration using simulated data in the "simulation/" folder within the Github repository. This directory also contains code and results for performance comparison between sn-spMF and other MF methods on simulated data.

2. In response to point 4 from Reviewer 1, you state that you use "a fully automated procedure that identifies parameter settings leading to the most stable decomposition as well as the most independent factors (see Methods, Brunet et al 2004, Wu et al 2016)". However, it is not clear how to achieve this with the code provided in GitHub. For example, the run_MF_train_coph.R script included in GitHub still requires manual specification of K, alpha and lambda parameters. Is your recommendation to perform grid search over these parameters? If yes, what are the plausible ranges of values that should be considered? It would be extremely useful to include these details to the worked-out example mentioned above (and perhaps the manuscript as well). It is fine if all of the steps cannot be run as a single script (e.g. due to computational requirements), but it is still important to spell out what the recommended flow of the analysis is.

These are excellent points. We have added a “Model Selection” section in the documentation in the Github repository (https://github.com/heyuan7676/ts_eQTLs) with the detailed process (which we attach at the end of this response as a Supplement for reviewer convenience). We have also included worked-out examples using both real eQTL data and simulated data in the Github repository. Finally, we have also added clarifications including setting the hyper-parameter search space to the section “Methods: Lower dimensional representation of eQTL effects: Model selection” in the main manuscript.

3. I think you could communicate more clearly that one of the advantages of your method is that you can naturally use different subsets of eQTLs for learning the factors (fine mapped variants or lead variants) and assigning eQTLs to factors (all significant gene-variant pairs). Initially I thought that you performed matrix factorisation on millions of correlated eQTL variants, but this is clearly not the case. This approach allows you to neatly learn the factors using independent genetic signals and subsequently calculate loadings for a much larger set of correlated variants (LD friends) to enable downstream applications (e.g. motif enrichment). Again, demonstration of this could be part of the worked-out example.

We thank the reviewer for the suggestion. Accordingly, to emphasize this point, we broke out a subsection called “Preprocessing and input data” in the manuscript, which provides details on how we select a small subset of likely causal variants with the strongest effects across tissues. The input data is much smaller than the full set of eQTL summary statistics for all variants. In addition, at the end of that section, we added the following text:

“sn-spMF is able to learn the underlying patterns from a subset of representative eQTL summary statistics. In our case, we restricted to credible set variants with the strongest signals across tissues, as described above. Other users may choose another representative subset of variants of interest based on their preferred methods for selecting likely causal variants or lead variants, but regardless, sn-spMF does not require summary statistics for every tested variant to learn relevant factors.”

In the demo eQTL data, we show this feature using a subset of data points to learn the factor matrix, and then map all data points to the learned factors.

Minor comments

4. The code used for performing TF motif enrichment analysis is not present in the GitHub repository. Similarly, the code showing how you ran the other matrix factorisation methods is not available. Since you mention in your response that you communicated with the authors of flashr to make sure that you ran their code correctly, I think would be very valuable to make these examples available for others to inspect. And I really mean inspect here – making such code executable by others is really not necessary and often infeasible, but readers who want to replicate your results should still be able to inspect the code that you used. If you wish, you could make it as a separate GitHub repository (e.g. “Extended Methods”) where you explicitly

state that this is code used in the manuscript and you are not providing any support for it. E.g. you only promise to respond to help requests for your novel sn-spMF method.

Thank you. We have incorporated the reviewer's suggestion by creating a folder in the Github repository called "Extended_Methods/Downstream_analysis/scripts", including scripts performing downstream analysis for the matrix factorization methods. The scripts include 1). Enrichment of ts-eQTLs and u-eQTLs in chromHMM status; 2). Gene set enrichment analysis for ts-eQTLs and u-eQTLs genes; 3). TF motif enrichment analysis for u-eQTLs and ts-eQTLs. We have also updated the scripts of running heuristic methods and following downstream analysis in "Extended_Methods/Heuristic_1" and "Extended_Methods/Heuristic_2". Scripts of running other matrix factorization methods are stored in the folder "Extended_Methods/Other_MF_methods", and scripts for downstream analysis are stored in "Extended_Methods/Downstream_analysis".

5. For the sentence, "Since it has previously been shown that inconsistent directions of effect for eQTLs will often arise from allelic heterogeneity rather than true sharing [20], we constrained factors to be nonnegative.", you might also consider citing (Casale et al, PLoS Genetics 2017). They provide a very nice demonstration of how apparent sign flips in context-specific eQTLs are also largely driven by allelic heterogeneity.

This is very helpful. We have added a reference to this publication.

Supplement

Documentation on "Model Selection". This is also available in the documentation on Github (https://github.com/heyuan7676/ts_eQTLs):

In the sn-spMF model, we need to set hyper-parameters including the rank of the decomposition (K) and the sparsity penalties (α , γ). We recommend searching for the hyper-parameters (K , α , γ) in two steps:

1. Narrow the sparsity penalty hyper-parameter search space

Exploring three hyper-parameters jointly can be computationally expensive, and many values would produce clearly implausible models. In order to choose a tractable, appropriate range of hyper-parameter settings to evaluate, we recommend running the method for a broad range of well-separated settings of sparsity penalty hyper-parameters, such as [1, 10, 100, 500], and a

wide range of K chosen by considering the number of tissues or experiments in the data. This coarse-grained search step can be evaluated using the following guidelines:

- a). the sparsity of the solutions being in accordance with user expectations for their domain - if the reported sparsity is far below the expected sparsity, the chosen penalty parameters may be too small.
- b). behavior of the factor matrix: if the number of utilized factors become much smaller than the initial number of factors to start with (ie. a lot of factors become empty, having no non-zero entries), it means that the penalty parameters are likely too stringent.

An example to perform this step is as below:

```
iterations=20
for K in 10 15 20
do
    for alpha in 1 10 100 500
    do
        for lambda in 1 10 100 500
        do
            bash sn_spMF/1_run_parameter_scope_search.sh ${K}
            ${alpha} ${lambda} ${iterations}
        done
    done
done
```

To collect the results from multiple runs, users can run the following command. The output will be saved in `output/choose_para_preliminary.txt`

```
Rscript sn_spMF/tune_parameters_preliminary.R -f choose_para_preliminary.txt
```

We ran the code above for demo data. When examining the output file `output/choose_para_preliminary.txt`, we observe that α and γ of either 1 or 10 result in a factor matrix with sparsity of 10% – 50%, which is lower than our expectation for this example (or a multi-tissue domain such as GTEx). On the other hand, α and γ of 500 result in too few utilized, non-zero factors (for example, $K = 10$, $\alpha = 500$, $\gamma = 1$ result in only around 6 non-zero factors). α and γ of 10 or 100 appear to give a balance between sparsity in the factor matrix and number of non-zero factors. $K = 10$ results in the majority of solutions having 10 used factors, while $K = 20$ results in the majority of solutions having nearly 20 used factors, after eliminating the models with sparsity penalties that are too stringent. This shows that searching between 10 and 20 for K is reasonable in this situation. Thus we proceed to perform grid search for α and γ in the range of 10 to 100, and K in the range of 10 to 20.

2. Refine the sparsity penalty hyper-parameter selection

Within a manageable search space for the hyper-parameters as selected above, we then suggest searching settings using finer granularity and evaluating the learned models for stability along with independence between factors. For example, run for α and γ in [10, 20, 30, ... 100] or finer, and run the model multiple times from random initializations (ie. 30 times). We recommend using a maximum number of iterations for 100 or less for each run. If the model does not converge within 100 iterations, it is probably because the penalty parameters are too small, which leads to very slow optimization steps. Larger penalty parameters are suggested in the case where the model does not converge within 100 iterations. An example to perform this step is as below.

```
iterations=100
for K in {10..20}
do
    for alpha in {1..10}
    do
        for lambda in {1..10}
        do
            a=$(( 10*alpha ))
            l=$(( 10*lambda ))
            run sn_spMF/2_choose_hyperparameters.sh ${K} ${a} ${l}
        done
    done
done
```

Within these chosen search spaces, we evaluated sn-spMF models for all combinations of K , α and γ using 1) a previously defined criterion of matrix factorization stability by Brunet et al.², and 2) independence of the learned factors, which represents adequate sparsity. Considering the stochastic nature of matrix factorization, Brunet et al. proposed a method looking for the most stable factorization result, and this method has been applied in various studies^{2,3}. We obtained the consensus matrix C after 30 runs with random initialization for each model. The values in C are between 0 to 1, representing the proportion of runs in which a pair of tissues are assigned to the same factor. Using the C matrix, we computed the cophenetic correlation which is used to measure the degree of dispersion for the C matrix. Higher cophenetic correlation indicates a more stable factor matrix. To collect the evaluation metrics, users can run the following command. The output will be saved in output/choose_para.txt.

```
Rscript sn_spMF/tune_parameters.R -f choose_para.txt
```

Based on the evaluation metrics, we performed the following selection steps:

- We first eliminated some settings of K . Here, for each observed mean number of learned, non-empty factors K' (which may be less than the input K), we aggregated across the different settings of γ and α and computed the median cophenetic correlation ².
- We eliminated from consideration any settings of K corresponding to a K' with a median cophenetic correlation < 0.9 . Next, among the remaining individual settings, we eliminated any cophenetic correlation < 0.9 .
- Last, among these apparently stable settings, we selected the final hyper-parameters based on the minimum Pearson correlation between pairs of factors, to encourage independent factors and a level of sparsity that matches independent signals in the data. Here, we computed Pearson correlation for each pair of factors, took the Frobenius norm of the pairwise correlation matrix, and averaged this across the 30 randomly initialized runs for the same setting.

In the demo data, We chose settings of K corresponding to a K' higher than 9, such that the corresponding median cophenetic correlation is above 0.9, and followed steps b) and c) to select the optimal model solution. The script is available in `sn_spMF/choose_paras_sn_spMF.ipynb`. A separate example of learning the hyper-parameters is provided in `simulation/choose_paras_sn_spMF_simulation.ipynb` on simulated data. Details can be found in `simulation/`.

References:

- [1]. Rawlings, J., Pantula, S., Dickey, D.: Applied Regression Analysis - A Research Tool: Second Edition. Springer, New York (1998)

[2]. Brunet, J.-P., Tamayo, P., R Golub, T., P Mesirov, J.: Metagenes and molecular pattern discovery using matrix factorization. Proceedings of the National Academy of Sciences. 101, 4164–9 (2004). doi:10.1073/pnas.0308531101

[3]. Wu, S., Joseph, A., S. Hammonds, A., E. Celniker, S., Yu, B., Frise, E.: Stability-driven nonnegative matrix factorization to interpret spatial gene expression and build local gene networks. Proceedings of the National Academy of Sciences 113, 201521171 (2016). doi:10.1073/pnas.1521171113